



# Satellite-based evaluation of AeroCom model bias in biomass burning regions

Qirui Zhong[1], Nick Schutgens[1], Guido van der Werf [1], Twan van Noije[2], Kostas Tsigaridis[3,4], Susanne E. Bauer[4,3], Tero Mielonen[5], Alf Kirkevåg[6], Øyvind Seland[6], Harri Kokkola[5], Ramiro Checa-Garcia[7], David Neubauer[8], Zak Kipling[9], Hitoshi Matsui[10], Paul Ginoux[11], Toshihiko Takemura[12], Philippe Le Sager[2], Samuel Rémy[13], Huisheng Bian[14,15], Mian Chin[15], Kai Zhang[16], Jialei Zhu[17], Svetlana G. Tsyro[6], Gabriele Curci[18,19], Anna Protonotariou[20], Ben Johnson[21], Joyce E. Penner[22], Nicolas Bellouin[23], Ragnhild B. Skeie[24], and Gunnar Myhre[24]

[1]Department of Earth Sciences, Vrije Universiteit, Amsterdam, The Netherlands.
[2]Royal Netherlands Meteorological Institute, De Bilt, the Netherlands.
[3]Center for Climate Systems Research, Columbia University, 2880 Broadway, New York, NY 10025, USA.
[4]NASA Goddard Institute for Space Studies, 2880 Broadway, New York, NY 10025, USA.
[5]Finnish Meteorological Institute, Kuopio, Finland.
[6]Norwegian Meteorological Institute, Oslo, Norway.
[7]Laboratoire des Sciences du Climat et de l'Environnement, IPSL, Gif-sur-Yvette, France.
[8]Institute for Atmospheric and Climate Science, ETH Zurich, Zurich, Switzerland.
[9]European Centre for Medium-Range Weather Forecasts, Reading, UK.
[10]Graduate School of Environmental Studies, Nagoya University, Nagoya, Japan.
[11]NOAA, Geophysical Fluid Dynamics Laboratory, Princeton, NJ, USA.
[12]Research Institute for Applied Mechanics, Kyushu University, Fukuoka, Japan.
[13]HYGEOS, Lille, France.
[14]University of Maryland, Baltimore County (UMBC), Baltimore, MD, USA.
[15]NASA Goddard Space Flight Center, Greenbelt, MD, USA.
[16]Pacific Northwest National Laboratory, Richland, WA, USA.
[17]Institute of Surface-Earth System Science, School of Earth System Science, Tianjin University, Tianjin 300072, China.
[18]Department of Physical and Chemical Sciences, University of L'Aquila, L'Aquila, Italy.
[19]Center of Excellence in Telesening of Environment and Model Prediction of Severe Events (CETEMPS), University of L'Aquila, L'Aquila (AQ), Italy.
[20]Department of Physics, University of Athens, Athens, Greece.
[21]Met Office, Exeter UK.
[22]Department of Climate and Space Sciences and Engineering, University of Michigan, Ann Arbor, MI, USA.
[23]Department of Meteorology, University of Reading, Reading, UK.
[24]Center for International Climate and Environmental Research-Oslo (CICERO), Oslo, Norway.

*Correspondence to*: Qirui Zhong (q.zhong@vu.nl)



**Abstract.** Global models are widely used to simulate biomass burning aerosols (BBA). Exhaustive evaluations on model representation of aerosol distributions and properties are fundamental to assess health and climate impacts of BBA. Here we conducted a comprehensive comparison of Aerosol Comparisons between Observation project (AeroCom) model simulations with satellite observations. A total of 59 runs by 18 models from three AeroCom Phase III experiments (i.e., Biomass Burning Emissions, CTRL16, and CTRL19) and 14 satellite products of aerosols were used in the study. Aerosol optical depth (AOD) at 550 nm was investigated during the fire season over three key fire regions reflecting different fire dynamics (i.e., deforestation-dominated Amazon, Southern Hemisphere Africa where savannas are the key source of emissions, and boreal forest burning on boreal North America). The 14 satellite products were first evaluated against AErosol RObotic NETwork (AERONET) observations, with large uncertainties found. But these uncertainties had small impacts on the model evaluation that was dominated by modeling bias. Through a comparison with Polarization and Directionality of the Earth's Reflectances (POLDER-GRASP) observations, we found that the modeled AOD values were biased by -93–152%, with most models showing significant underestimations even for the state-of-art aerosol modeling techniques (i.e., CTRL19). By scaling up BBA emissions, the negative biases in modeled AOD were significantly mitigated, although it yielded only negligible improvements in the correlation between models and observations, and the spatial and temporal variations of AOD biases did not change much. For models in CTRL16 and CTRL19, the large diversity in modeled AOD was in almost equal measures caused by diversity in emissions, lifetime, and mass extinction coefficient (MEC). We found that in the AEROCOM ensemble, BBA lifetime correlated significantly with particle deposition (as expected) and in turn correlated strongly with precipitation. Additional analysis based on Cloud-Aerosol LIdar with Orthogonal Polarization (CALIOP) aerosol profiles suggested that the altitude of the aerosol layer in the current models was generally too low, which also contributed to the bias in modeled lifetime. Modeled MECs exhibited significant correlations with the Ångström Exponent (AE, an indicator of particle size). Comparisons with the POLDER-GRASP observed AE suggested that the models tended to overestimate AE (underestimated particle size), indicating a possible underestimation of MECs in models. The hygroscopic growth in most models generally agreed with observations and might not explain the overall underestimation of modeled AOD. Our results imply that current global models comprise biases in important aerosol processes for BBA (e.g., emissions, removal, and optical properties) that remain to be addressed in future research.





## 1 Introduction

Biomass burning (BB) injects large amounts of aerosols into the atmosphere every year. It is estimated that BB is responsible

for 26–73% and 27–41% of global organic carbon (OC) and black carbon (BC) emissions, respectively (Bond, 2004; Andrea and Rosenfeld, 2008; Wiedinmyer et al., 2011; Wang et al., 2014; Huang et al., 2015). As a result, BB aerosol (BBA) has a considerable impact on human health and the global climate. For example, numerous studies have shown that exposure to BBA can cause cardiovascular diseases and subsequently lead to premature death (Johnston et al., 2012; Lelieveld et al., 2015). In addition, BBA can also alter the global and regional energy budgets by interacting with solar radiation directly, and

indirectly by modifying the lifetime and albedo of cloud through their role as cloud condensation nuclei and ice-nucleating particles (Engelhart et al., 2012; Jahl et al., 2021). On a global scale, assessments of these health and climate impacts rely directly or indirectly on model simulations regarding BBA's distributions, compositions, and properties (Martins et al., 2009; Lin et al., 2014; Dong et al., 2019).

One of the frequently used variables to define model representation for BBA is aerosol optical depth (AOD) which

depends on both aerosol abundance and optical properties in the atmosphere. Previous studies have reported that global models produced substantial underestimations of AOD over BB regions with highly varying extent despite using different emission inventories (Kaiser et al., 2012; Veira et al., 2015; Johnson et al., 2016; Reddington et al., 2016; Mallet et al., 2021). For example, Kaiser et al (2012) showed the global Monitoring Atmospheric Composition and Change (MACC) aerosol model driven by emissions from the Global Fire Assimilation System (GFAS) underestimated AOD by a factor of 2–

4 for BBA. While Johnson et al (2016) found that the AOD was underestimated by a factor of 1.6–2 in the simulations by Hadley Centre Global Environment Model version 2 and 3 (HadGEM2 and HadGEM3) based on the Global Fire Emission Database version 3 (GFED3). The systematic underestimation of AOD in global models suggests a potential negative bias in current BB emission inventories (Reddington et al., 2016). Several factors could contribute to producing such bias in emission inventories based on either satellite-detected burned areas (e.g., van der Werf et al., 2017) or fire radiative power

(FRP, e.g., Ichoku and Ellison, 2014). The burned-area-based emission inventories comprise uncertainties in satellite detection of burned areas and fuel load (Randerson et al., 2012; Andela et al., 2016), while FRP-based emission datasets are largely affected by the translation of FRP into rates of biomass combustion (Kaiser et al., 2012). In addition, both emission datasets rely on uncertain emission factors converting burned biomass to trace gas or aerosol emissions (Stockwell et al., 2015). Moreover, when these emission inventories are used to run models, the OC emissions will be converted to emissions

of organic aerosols (OA) based on the assumed OA/OC ratio which differs extensively among models (Gliß et al., 2021). It's thus expected to see large diversities in simulated AOD from models driven by varying BBA emission inventories.

In addition to emissions, model performance for simulating BBA also depends on model configurations. This has been reported for individual models. Reddington et al (2019) showed that increasing the aerosol hygroscopicity can reduce AOD



errors simulated by Global Model for Aerosol Processes (GLOMAP) over tropical BB regions. A similar impact of
hygroscopicity was also observed in Johnson et al (2016) by comparing the modeled AOD errors between two aerosol
schemes in HadGEM3 model. Schill et al (2020) found that the large BBA biases in the remote troposphere could be
eliminated by increasing wet removal strength. Additional configurations that can alter model performance include, for
example, model resolution (Bian et al., 2009), particle size distribution (Chin et al., 2009), complex refractive index (Brown
et al., 2021), aerosol lifetime (Bauer et al., 2013), and aerosol mixing state (Cappa et al., 2012; Brown et al., 2021). With
different assumptions, methodologies, and parameterizations selected for aerosol processes in models, model evaluations can
be very different even when the same emission inventory is used.

Apart from the issues in emissions and model configurations, the uncertainty in observations is another factor affecting
model evaluations. AErosol RObotic NETwork (AERONET) is frequently used as a solid observation dataset for aerosols
(Tombette et al., 2008; Smirnov et al., 2011). However, the AERONET network is not particularly well aligned with BBA
regions and available observations are limited (e.g., in Africa, Siberia). Over specific BB regions, flight campaign
measurements are applied to be compared with models for certain periods (e.g., Myhre et al., 2003; Johnson et al., 2016).
But the temporal coverage of these campaigns is limited given the large inter-annual variability of fires (van der Werf et al.,
2017), and the observations suffers from uncertainties due to sampling instruments (Pistone et al., 2019). In comparison,
satellite datasets provide more continuous observations in space and time. Unfortunately, satellite remote sensing, conducted
by either a polar-orbiting or geo-stationary satellite, suffers from a series of uncertainties and noise that can originate from
radiance calibration, cloud screening, the effects of strong surface reflection, and the variation in aerosol particle sizes and
components (Li et al., 2009; Schutgens et al., 2020; Falah et al., 2021). As a result, the satellite retrieved AOD displays
significant variations. For example, Schutgens et al (2020) found that the diversities of individual satellite products can reach
up to 100% on regional scales. It is therefore necessary to understand the uncertainties in the satellite products prior to the
model validation.

To better quantify and interpret the model bias of BBA, we conducted a comprehensive inter-comparison between
various global models and observations. The aim of this work is to provide a satellite-based assessment of global models in
representing BBA and to see if models have been improved regarding to knowing issues for BBA models for more than ten
years. This study focusses on AOD at 550 nm–a basic optical property used to measure the abundance of aerosols in the
atmosphere–during fire seasons. A model ensemble was built from three phase-III experiments of the Aerosol Comparisons
between Observation (AeroCom) project. Such a comparison between models and satellite observation ensembles will
provide more robust results than individual comparisons, and the spread of individual models allows an in-depth
interpretation of the modeled diversities. Additional modeled variables and observations (e.g., total emissions, aerosol load,
precipitation, plume height, Ångström Exponent, hygroscopic growth) were also used to further aid in the interpretation.



Prior to the model validation, we assessed a total of 14 satellite products to identify the possible uncertainties induced by observations of AOD. The paper is organized as follows. The details of the methodology and data sources are presented in Section 2. Section 3 evaluates satellite observation uncertainties over the selected fire regions and their impacts on model validations. Section 4 quantifies the model bias in AOD. Section 5 presents the diversity in modeled AOD which is further interpreted through three aspects of the modeling processes.

## 2 Data and Methods

### 2.1 Models and variables

This study evaluated the AOD at 550 nm simulated by models from three AeroCom Phase-III experiments: the biomass burning emissions experiment (BBE), control experiment 2016 (CTRL16), and control experiment 2019 (CTRL19). **Table 1** provides an overview of all the models involved in our evaluation, more details are provided in the appendix questionnaire

and listed references. A total of 18 different models were investigated in our study, and some models participated in multi experiments with different versions (**Table 1**). The general settings of the three experiments were as follows.

The aim of the BBE was to quantify the impact of BBA emissions on AOD simulations. All the participating models presented simulations for the year 2008 using the prescribed BB emission input (GFED3). In addition, simulations with scaling factors of 0, 0.5, 2, and 5 (referred to as BBE0, BBE0.5, BBE2, BBE5) adapted to GFED3 emissions were also

provided. These scaling factors were based on a preliminary simulation by the Goddard Chemistry Aerosol Radiation and Transport (GOCART) aerosol model, which found that using default GFED3 emissions would lead to AOD underestimations over most fire regions (Petrenko et al., 2012). The perturbations in emissions would allow a quantitative analysis of the AOD-emission response.

The models in CTRL16 adopted the standard diagnostics and presented simulations for 2006, 2008, and 2010. The

modelers were advised to nudge the meteorology to (or drive the models by) their preferred datasets (see **Table 1**). The standard outputs mainly included 2-D fields at a monthly frequency, which were extended by several other experiments launched subsequently (e.g., the remote sensing experiment). High-frequency (3-h) AOD data together with other information (e.g., 3-D fields of the AOD) are currently available. In this study, we examined 12 models with an AOD output at a 3-h frequency for 2006, 2008, and 2010.

The state-of-art of aerosol modeling for 1850 (pre-industrial era) and 2010 (present day) was assembled in CTRL19. All models were nudged to (or driven by) a fixed sea surface temperature and 2010 meteorology using different data sources (see **Table 1**). Emissions from Coupled Model Intercomparison Project Phase 6 (CMIP6) were used when applicable. The





model AOD was output at a daily or monthly frequency. In this study, we selected 12 models that provided a daily output for 2010.

In addition to AOD, additional variables from the models were used to interpret model diversity when available. These additional variables included emissions, total deposition (both dry and wet deposition), aerosol column load (with aerosol species resolved), the vertical profile of the extinction coefficient (EC), precipitation, and Ångström Exponent (AE, which was calculated using the AOD at 440 and 550 nm, the AE-based interpolation was adopted if AOD at 440 nm was not available for some models).

We also prepared a questionnaire filled by modelers to acquire information on the model configuration details (see **Appendix**). Information was collected for models in CTRL16 and CTRL19.

## 2.2 Fire regions

Based on the models considered, three key BB regions were selected in this study: Amazon (AMAZ), Southern Hemisphere Africa (SHAF), and boreal North America (BONA). **Figure 1** shows the domains of these three regions and the
corresponding OC emissions from BB. In terms of their aerosol emission, different fire types could be identified in each region. The BB emissions in AMAZ were dominated by tropical forest fires and deforestation, whereas emission from savanna grassland fires was the major source in SHAF. In BONA, BB aerosols were mainly emitted from boreal forest fires. Regions with agricultural waste burning or temperate forest fires were not considered due to their small contribution on a global scale (van der Werf et al., 2010). Using the satellite observation of AOD, we defined the fire seasons (dry seasons)
over the three regions (see **Figure 1b**) that were investigated in this study.

## 2.3 Observation data

A total of 14 satellite AOD datasets were used in this study. **Table 2** provides an overview of the datasets. The AOD data at 550 nm wavelength were obtained by either direct retrieval or interpolation/extrapolation from the AOD at nearby wavelengths.

The ground-based remote sensing data were taken from AERONET DirectSun L2 v3 (Dubovik et al., 2000). The locations of the AERONET sites within the three fire regions are shown in **Figure 1a**. Given that the sparse distribution of AERONET sites results in poor spatial data coverage, especially in SHAF and BONA, we mainly used the AERONET data to evaluate the satellite datasets, while model validations relied on satellite data.

For the vertical profiles, we used the Cloud-Aerosol LIdar with Orthogonal Polarization (CALIOP) L2 Layer 5 km v4.20
product. The EC data at 532 nm were compared with models (at 550 nm) where the vertical data were available. For CALIOP data, we only considered columns that had at least one aerosol retrieval based on the cloud-aerosol discrimination



(CAD) scores (CAD < -20) (Watson-Parris et al., 2018). Columns with extreme CAD scores (< -100) were also excluded because they might have been the result of bad shots (Watson-Parris et al., 2018). To ensure data quality, we only used the most reliable retrievals that had extinction quality control (QC) flags of 0, 1, 2, 16, or 18. In addition to the direct comparison of vertical extinction profiles, we calculated the weighted mean plume extinction height (PEH) based on the vertical EC and layer height ($h_i$) for the aerosol layers below 6 km (Koffi et al., 2016), as shown by Eq. (1):

$$PEH = \frac{\sum EC_i h_i}{\sum EC_i} \tag{1}$$

In addition, we evaluated the modeled precipitation as it is the cause of a major deposition process. The precipitation data were taken from the Global Precipitation Climatology Project (GPCP), which incorporates precipitation from low-orbit satellite data, geosynchronous satellite data, and surface raindrop observations (Adler et al., 2003).

### 2.4 Data analysis

To avoid sampling issues, we conducted strict collocations before the data were evaluated (Schutgens et al., 2016a; b). Both model and observation data were firstly re-gridded into the 1° × 1° spatial grid-boxes. The temporal resolution was aggregated into 3-h or daily intervals according to the model output frequency (see **Table 1**). For the satellite validation against AERONET, we compared satellite data with AERONET at the resolution of 1° × 1° × 3-h. Specially, the plume height in models was validated against CALIOP on monthly basis since CTRL19 models only provided data at such a resolution. Vertically, the CALIOP data were aggregated into 100-m intervals and all the extinction profiles from models were linearly interpolated into the same resolution for validation.

The data aggregation and collocation were processed via a command-line tool called Community Intercomparison Suite (CIS, Watson-Parris, et al., 2016). To quantitatively evaluate the model performance and satellite observation uncertainties, we utilized Taylor diagrams to present the statistics, including the Pearson correlation coefficient (R), standard deviation (SD) and centered root mean square error (CRMSE) (Taylor, 2001). Taylor diagrams are presented in polar coordinates with the polar axis showing the SD of evaluated data and cosine of the polar angle showing the R-value between evaluated and 'reference' data. The distance between the evaluated and 'reference' data shows the CRMSE according to the law of cosines. Both evaluated and 'reference' data were normalized by the SD of 'reference' data so that the 'reference' was always located at [1, 0] (see **Figure 2a** for an example). A Taylor diagram is a convenient way to visualize the performance of models or observations versus a reference data set. However, bias is not shown by Taylor diagrams, and we accompanied each Taylor diagram with a plot showing the normalized mean bias (NMB, defined as the mean bias divided by the mean value of observation) to provide a comprehensive evaluation.



## 3 Evaluation of satellite products

A large number of satellite AOD datasets have become available, and it is important to use the dataset that can adequately serve the specific research goal. In light of the uncertainties in satellite observations, we evaluated individual satellite datasets against AERONET observations before model validation in the three fire regions. The evaluation was only conducted for data during the fire seasons, and most observations were collected over AMAZ.

**Figure 2** shows the evaluation of 14 satellite datasets against AERONET observations for the three fire regions during the fire season. The data points in the Taylor diagram were normalized by AERONET data with a different sampling for each product (**Figure 2a**), while NMBs are shown in the scatter diagram (**Figure 2b**). All the satellite datasets agreed with AERONET observations over AMAZ better than the other two regions, with stronger correlations (R = 0.85–0.95) and lower normalized CRMSE (< 0.5). For AMAZ, all the datasets had similar correlations and CRMESs but very different biases. The POLDER-GRASP dataset and two algorithms adopted to Moderate Resolution Imaging Spectroradiometer (MODIS) data (BAR and DarkTarget) tended to overestimate AOD (3–13%), while the others resulted in underestimations (-1–-20%). Unlike in AMAZ, individual satellite products agreed less well with AERONET and there were strong variations within each of them over SHAF and BONA (R = 0.31–0.91, CRMSE = 0.51–1.71). All products except Aqua-MODIS-BAR underestimated AOD over SHAF (-7–-73%), whereas most products overestimated AOD over BONA by up to +73%. Both the spatial and temporal data coverages in BONA and SHAF were much lower than in AMAZ, which suggested that there might be more cloud contamination issues in the two regions and higher biases could be expected (Schutgens et al., 2020). Generally, we found that MODIS products agreed well with the AERONET data, although details vary by the retrieval algorithm. For example, the MODIS-BAR products were the best in AMAZ and SHAF, while the MODIS-MAIAC product was better than the others in BONA. From the perspective of bias, we found that the variations among satellite products were affected more by the algorithm than the instrument, which was related to the amount of spectral information used in the retrieval. For example, the data spread of the four instruments that adopted the DeepBlue algorithm (i.e., Aqua-MODIS-DB, Terra-MODIS-DB, AVHRR-DB, and SeaWiFS-DB) was smaller than that for the MODIS products that used four different algorithms (i.e., BAR, DB, DT, and MAIAC) for all three regions.

It should be noted that the evaluation was affected by representation issues. As shown by **Figure 1**, there were more AERONET sites located in fire areas in AMAZ. While in SHAF, the AERONET sites were far from the fire emission sites and the downwind area and only captured a small part of the BB aerosol signals. In BONA, the temporal coverages of both AERONET and satellites were poor. Due to the stratocumulus and low broken cumulus cloud contamination, satellite retrievals of AOD were enhanced, which could lead to unexpected overestimations when compared with the ground-based observations over BONA (Toth et al., 2013).





In this study, we proposed to use POLDER-GRASP to evaluate all the models. AOD from POLDER-GRASP has been validated in a previous study which suggests POLDER-GRASP is superior to other products globally (Schutgens et al., 2021). The AE data has also been validated before, showing a good agreement with AERONET (Chen et al., 2020). In our study, we also investigated how the observation uncertainties mentioned above may affect model validations, which were indicated by the interquartile ranges of the R, CRMSE, and NMB based on validations using different satellite products. The

interquartile values were further compared with the statistics (i.e., R, CRMSE, and NMB) when using POLDER-GRASP to show the uncertainty range when using the specific dataset. Before calculating the difference in model validation (i.e., R, CRMSE, NMB) due to different satellite products, each model was collocated with satellite products either individually (i.e., all the models were collocated with the different sampling of each satellite product, see **Figure 3a-c**) or synchronously (i.e., the model data were collocated with the same sampling where all satellite products could provide data, see **Figure 3d-f**). In

the latter case, only products that had a similar overpass time with POLDER-GRASP were considered (i.e., with an overpass time in the afternoon, excluding datasets onboard Terra and ENVISAT). For comparison, the uncertainty ranges of 25%, 50%, and 100% for the relative uncertainties to POLDER-GRASP were also shown. We found that for R and CRMSE using an individual collocation (**Figure 3a-c**), the uncertainties due to the different satellite products were generally lower than 25%, indicating a small impact when using different satellite datasets. The impact on CRMSE was slightly stronger than that

on R, which suggested more agreements among the different satellite products (or better performance) in capturing the spatiotemporal trends of AOD than the magnitude. In the case of NMB, the impacts of different were large only when the modeled NMB was small (<20%). The majority of simulations had an NMB higher than ± 40%, suggesting the uncertainties among the different satellite products were less important for NMB and the modeled bias was dominated by the biases in the model instead of the difference in satellite products. For the synchronous collocation which eliminated the sampling

differences (**Figure 3d-f**), similar results were obtained with even much smaller satellite uncertainties. In this case, all the satellite products were collocated, which greatly reduced the frequency of cloud contamination issues and provided more reliable results. Due to the synchronous collocation, a large portion of the original observations was filtered and statistical noise may stand out. We then conducted a 10,000-time bootstrap sampling with replacement to examine the potential effects of such noise. In each time, we randomly excluded 20% of the data to test the robustness of our evaluations. The coefficient

of variation for the satellite observation uncertainties from the 10,000-time bootstrap sampling was 1–10% for R, 1–12% for CRMSE, and 3–27% for NMB. For the stronger variation in NMB, over 85% of simulations were subject to an NMB variation of less than 10%, suggesting very robust results for the above analysis. All this indicated that although there were different errors in these satellite products, only a small part (accounting for < 25% of the modeled errors) could be expected to affect the model validation. Given the small impacts, we decided to validate models against POLDER-GRASP product for

both AOD and AE, which provided a degree of consistency for the whole research.





However, the small validation impacts of using different satellite products were only found in our validations over BB regions during BB seasons. Such a conclusion cannot be directly applied to other areas/periods. For example, for the same fire regions outside the fire seasons, we found that the uncertainties due to different satellite products could be as high as 50% in most cases (not shown).

## 4. Evaluation of AeroCom models

We then evaluated AOD in AeroCom models in three experiments using the POLDER-GRASP product. All model data were collocated with POLDER-GRASP sampling. The model evaluation is shown in **Figure 4** via Taylor diagrams and bias plots. The R-values ranged from 0.1 (INCA over BONA in BBE) to 0.78 (ECMWF-IFS over SHAF in CTRL19) for all models and regions, with a median value of 0.63. Over 80% of the model simulations had an R-value higher than 0.5, but only 24% of simulations had correlations stronger than 0.70, suggesting a generally moderate capability for capturing the spatiotemporal variation in aerosol data. For CRMSE, the modeled variation (defined as the inter-quantile range divided by the median value, 51%) was stronger than that for R (22%), indicating a higher modeled disparity of the AOD magnitude than the spatiotemporal trends. Based on an analysis of variance for R and CRMSE (**Figure 4a-c**), we found that the models showed similar performance over the three regions as there was no significant difference found. The median NMB of models (**Figure 4d**) for AMAZ, SHAF, and BONA were -28% (-6%–-54% as inter-quartile), -54% (-30%–-63%), and -54% (-46%– -57%), respectively. Models produced significantly smaller NMB over AMAZ than over the other two regions, though the inter-model variation was also found to be the highest among the three regions. More than half of the simulations showed an underestimation of AOD by a factor of > 2, consistent with previous studies (e.g., Kaiser et al., 2012; Veira et al., 2015; Johnson et al., 2016).

In addition to the overall model evaluations, we also evaluated the modeled temporal (time series for the whole fire regions) and spatial patterns (temporal averages for individual grid-boxes during fire seasons). In **Figure 5**, we compared the temporal and spatial correlations of modeled AOD with observations. Most models showed similar temporal and spatial correlations ranging from 0.6 to 0.9 which were slightly higher than the overall correlations shown in **Figure 4** due to data averaging. Both the spatial and temporal correlations in most models clustered in this range, which partly explained the similarity of the overall correlations mentioned above. We found there was no significant difference between the temporal and spatial correlations in individual models from the three experiments. Although the AOD errors differed substantially per model, the spatial and temporal variation among models tended to be small. For most models, we did not observe significant improvements in spatial and temporal correlations following the time sequence from BBE to CTRL16 to CTRL19. We also compared the variations of temporal and spatial AOD biases AOD biases, as shown in **Figure 6**. Here the variations were





defined as the ratio of interquartile to median values of the time series (temporal variations) or spatial averages (spatial variations) of absolute modeled AOD bias. The spatial variations were significantly smaller than temporal variations for all three experiments, suggesting the different temporal evolution of AOD biases was the leading cause of the large NMB diversity in **Figure 4**. It partly suggested that current emission inventories had a better representation of BBA emissions over space than over time.

Since the modeled AOD bias is strongly affected by input emissions (Kaiser et al., 2012; Johnson et al, 2016), we also investigated the model response to the changes in emissions based on BBE experiment. This scaling-up procedure has been used to fix overall AOD errors for BBA in previous studies (e.g., Kaiser et al., 2012; Johnson et al, 2016; Veira et al. 2015). **Figure 7** shows the evaluation of these models for R, CRMSE, and NMB. As expected, NMB increased monotonously with the increase of emissions. Most models would produce significant positive bias when the scaling factors to GFED3 reached

5, but more than half of models still underestimated AOD when BBA emissions were doubled. Such trends were also found for CRMSE with a much weaker sensitivity. Similar phenomena were also found in the other two experiments. For example, we found the ECHAM-HAM model agreed well with observation in CTRL16 experiment which used $3.4 \times$ GFAS emissions, while it produced large underestimation when the CMIP6 emissions (much lower than $3.4 \times$ GFAS) were used (see **Figure 4d**). Given the metrics of CRMSE and NMB, the ensemble of models in BBE experiment showed the best

agreement with observations when the emissions were scaled by a factor of 2. This systematic response of modeled bias also suggested a possible underestimation of emissions in the applied inventory (GFED3). However, correlations in most models did not improve along with the increased emissions, since there was no further spatiotemporal information added into the emissions.

The modeled AOD bias during fire seasons could be due to both BBA and background sources (e.g., anthropogenic,

biogenic, dust, and sea salt aerosols). However, it is difficult to isolate BBA errors from the background based on existing simulations. Since we found that most models underestimated AOD in the BBE1 simulations, it was not possible to determine the real BBA impacts by comparing BBE1 and BBE0 simulations. Instead, we compared the collocated BBE0 AOD (background) with POLDER-GRASP observation during fire seasons. The modeled AOD in BBE0 varied substantially by a factor of 9 in the three regions. Compared with observations, the background averagely accounted for only

14%, 12%, and 11% of total AOD over AMAZ, SHAF, and BONA, respectively. We also compared the modeled AOD biases during non-fire seasons with those during fire seasons, with the former showing much smaller magnitude compared with the latter (0.04 vs 0.35, for the absolute mean bias). This analysis supports the notion that AOD bias over the fire regions was dominated by the BBA rather than background sources.





## 5. Model diversity and its interpretation

In this section, we explore the diversities of AOD which lead directly to the differences in bias discussed above (see **Figure 4**). Understanding the model diversities and the drivers would improve our knowledge of model bias, which will enable further development of the models. Our strategy is to first evaluate the variation in modeled AOD and the possible causes that could lead to such model variability, and then compare those causes with observations to understand the model variability and therefore bias. Unless stated otherwise, data in this section are presented as area averages for the whole fire

season based on the raw model outputs without any collocation. The aim is to determine the general drivers of variation in AOD for model ensembles rather than individual models, although evaluations for specific models are also presented where sufficient information is available.

The diversities of AOD were decomposed into three factors, i.e., total aerosol emissions, aerosol lifetime, and the MEC, as described by the following function:

$$AOD = Emission \times Lifetime \times MEC \tag{2}$$

where emission indicates the total emissions of OA (including secondary organic aerosols which were treated as emitted aerosols given the fast transformation), BC, sulfur dioxide ($SO_2$), sulfate (SU), mineral dust (DU), and sea salt (SS) within the fire regions; lifetime is defined as the average total aerosol load divided by total emissions within the fire regions; and the MEC is defined as AOD divided by total aerosol load, which is strongly associated with the modeled aerosol optical

properties (e.g., size distribution, refractive index, hygroscopicity, etc.). Emissions, aerosol load, and AOD were first calculated as regional and seasonal averages so that the lifetime and MEC were determined on a seasonal level for the focused regions. Note that the definition of lifetime in this study is different from the usual one as we are considering open systems. However, the time scale here called lifetime is still determined by the same relevant process (e.g., depositions). This is discussed in detail in Sec. 5.1.

**Figure 8** shows the diversities of the three factors. The slope of the line between each dot and the origin indicates the aerosol lifetime (**Figure 8a**) and MEC (**Figure 8b**) for a specific model averaged for the whole fire season, respectively. The emissions varied by a factor of 10 among the models. Such large deviations resulted from different emission inventories (mainly for CTRL16 models) and the different schemes for estimating non-BB aerosols (e.g., dust, biogenic sources). For CTRL19 experiment with its prescribed emission inventory (CMIP6), the input emissions were altered mainly by the

different OA/OC ratios, and to a lesser extent by the different mechanisms of DU production and biogenic sources. For example, the OA/OC ratios were set as 1.4, 1.8, and 2.6 in ECHAM-HAM, GEOS, and SPRINTARS, leading to emissions being 34% and 88% higher in the latter two models, respectively. The difference in these ratios is a consequence of the different assumptions regarding the oxidation of freshly emitted OC. The widely used ratio of 1.4 was established based on





field measurements over urban regions (Turpin and Lim, 2001) and was therefore more representative for anthropogenic OC

emissions. More recent investigations of the BB plume have suggested that the oxidation levels are higher for both fresh and aged BB OC particles (Aiken et al., 2008; Brito et al., 2014; Tiitta et al., 2014). Increasing the OA/OC ratio can directly lead to an elevated AOD in models and a higher ratio than 1.4 has been suggested for BB aerosols in some previous studies (e.g., Reid et al., 2005; Aiken, et al., 2008; Johnson et al., 2016). Omitting GISS-MATRIX and GISS-OMA which produced an unexpected positive AOD bias (see **Figure 4**), we found that the modeled NMB generally decreased with an increase in the

OA/OC ratio for CTRL19 models. For example, the average NMB of AOD for the model group that used the ratio of 1.4 (i.e., CAM5-ATRAS, ECHAM-HAM, and ECHAM-SALSA) was -61%, whereas the value was only -22% for the group using a ratio of 2.6 (i.e., CAM-Oslo, NorESM, OsloCTM, and SPRINTARS). The NMB of the models using a ratio of 1.6– 1.8 was within an intermediate range (-43% to -46%). This shows the importance of determining realistic values of the OA/OC ratio. However, it does not necessarily mean that higher OA/OC ratios can address the underestimated AOD. For

example, both SPRINTARS and OsloCTM produced significant overestimations in AMAZ using an OA/OC ratio of 2.6 that was higher than many in-situ observations (e.g., Brito et al., 2014; Zheng et al., 2017).

When all three fire regions were considered simultaneously, there was a general linear response of the aerosol load to aerosol emissions, and of AOD to aerosol loads. Nevertheless, significant diversities in lifetime and MEC were found. For the three regions, the relative variation (i.e., interquartile value divided by the median value) was found to be the lowest for

the MEC (49%, 41%, and 40% for AMAZ, SHAF, and BONA, respectively), moderate for aerosol lifetime (62%, 49%, and 26%, respectively), and highest for emissions (62%, 95%, and 64%, respectively). For the aerosol lifetime and MEC which were mainly affected by other model aspects than emissions, there was no significant difference found among the three fire regions for the same model. HadGEM3 over BONA presents an outlier case for lifetime which is probably related to high local DU emissions. We also noticed that the DU emission in HadGEM3 covered a much wider area than in the other models

due to the use of different mechanisms (Woodward, 2001; Mulcahy et al., 2020).

The contributions of aerosol emissions, aerosol lifetime, and the MEC (which were found to be statistically independent on each other) to the overall variation in AOD were evaluated. We used Eq. 2 to investigate such contributions. In the case of the AOD variation induced by emissions, we calculated the AOD variation (i.e., the standard deviation) over all modeled emissions and a random combination of aerosol lifetime and MEC values from the model ensemble. This calculation was

repeated for all the combinations of aerosol lifetime and the MEC, and the variation in AOD attributable to emissions was then quantified as the average value of all the standard deviations. Similar calculations were also applied to aerosol lifetime and MEC values. It was estimated that aerosol emissions, aerosol lifetime, and the MEC accounted for 38%, 33%, and 29% of the variation in AOD, respectively, suggesting only small differences in determining the overall variation, although emissions might be slightly more important than aerosol lifetime and the MEC. We also applied this evaluation to individual





fire regions and similar conclusions were obtained. This suggests that reducing the uncertainties associated with emissions
uncertainties might have only a moderate impact on the accuracy of the BBA simulations, and uncertainties in lifetime and
MEC should also be considered.

## 5.1 Diversity of aerosol lifetime

In this section, we discuss the potential factors that contribute to the diversity of aerosol lifetime. Because we focused on

three separate open systems, we described the aerosol budget of each region as a simple box model, as shown by Eq. (3):

$$\frac{dB}{dt} = E - D + I - O + P - L \rightarrow \frac{E}{B} = \frac{D}{B} - \frac{I-O+P-L}{B} \tag{3}$$

where $B$, $E$, $D$, $I$, $O$, $P$, and $L$ indicate the average of total aerosol burden, emission, deposition, inflow, outflow, chemical
production, and chemical loss of a focused region. For a closed system and a steady state without chemistry $I = O = P = L = 0$
and a lifetime can be defined as $E/B = D/B$. For an open system, steady-state and with on-going chemistry, E/B does not

equal to D/B but both still are time-scales defining the system. Here we show that for these fire regions, $E/B$ correlates with
$D/B$. **Figure 9** displays the linear dependence of the modeled aerosol lifetime on the time scale of total deposition and all
other processes. For most models, the reciprocal of aerosol lifetime ($E/B$) responded linearly to the time scale of deposition
($D/B$), except for INCA from CTRL16. This suggests that the difference in deposition is a leading contributor to the
variation in aerosol lifetime. HadGEM3 simulations in BONA (the outliers of the aerosol lifetime trend in **Figure 8**) still

followed the same linear trend, confirming that the short aerosol lifetime is a direct result of the strong deposition of coarse
mineral dust. For INCA, the simulated aerosol load was much lower than other models, and the modeled aerosol
composition was very different with OA contributing less than 20% of the total aerosol load. As a result, the coarse mode
aerosols dominated the total aerosol composition, resulting in a relatively short aerosol lifetime. When the INCA model was
omitted, the correlation between the reciprocal of aerosol lifetime and the deposition timescale was 0.95, suggesting that

90% of the modeled variation in aerosol lifetime could be explained by deposition. The variation in regional transport and
the chemical budget together only contributed around 10% of the variation in aerosol lifetime and was therefore much less
important to the overall difference in AOD. The timescale of the total deposition had a variation of 72% (i.e., the
interquartile value divided by the median) which was slightly higher than the aerosol lifetime (62%).

The modeled deposition was primarily a consequence of wet deposition (61% of the total deposition on average) even

during the dry fire season. The modeled wet deposition, which occurred mainly due to below-cloud scavenging (Andronache
2003; Zhang et al., 2004), was related to the size distribution of aerosols and raindrops as well as the precipitation intensity
(Seinfeld and Pandis, 2006). **Figure 10** compares the modeled timescale of total deposition and precipitation strength. Note
that not all models provided both deposition and precipitation outputs, the conclusion of the evaluation may need to be re-





examined when more data becomes available in the future. The modeled precipitation differed among the models by factors
of 3.8, 13.6, and 2.2 for AMAZ, SHAF, and BONA, indicating a substantial model discrepancy. When all regions were
considered, there is a significant positive correlation between the modeled precipitation and the timescale of total deposition.
For comparison, we also compared models with GPCP data. GEOS, SPRINTARS, and TM5 were among the models with an
overestimated precipitation in all three fire regions, which suggested systematic errors in the modeled lifetime. On a regional
basis, models exhibited large regional variations. Almost all models tended to overestimate the precipitation over BONA by
up to 69% (ECHAM-HAM from CTRL19), which might partly explain the underestimated AOD in this region. There were
large disparities in precipitation simulation over AMAZ, ranging from -21% to 130%. In contrast, we found that most
models underestimated both AOD and precipitation in SHAF, suggesting other important sources of AOD bias in addition to
precipitation. However, we did not observe a clear dependence of AOD biases on precipitation biases. For example, a bias of
6% and -9% were found for precipitation and AOD, respectively, over AMAZ in CAM5-ATRAS from CTRL19, whereas
the corresponding AOD biases were 14% and -86% over BONA. This suggests that other factors than precipitation affect
AOD biases significantly.

In addition to precipitation, we also examined the impacts of aerosol plume height on the aerosol lifetime. **Figure 11a**
compares the modeled plume height (as represented by PEH) and aerosol lifetime. Based on the limited number of models
with data available, there was a generally increasing trend in the aerosol lifetime as the plume height increased ($r = 0.65$)
except for one outlier (IMPACT over BONA), suggesting that plume height could also affect the modeled aerosol lifetime.
Generally, the modeled PEH varied by a factor of 4, partly due to the model assumption in the fire injection height for BB
emissions. For example, ECHAM-HAM and ECHAM-SALSA, which allowed 25% of BB aerosol emissions to be emitted
above the planetary boundary layer (PBL), generally had a higher plume height than models that distributed emissions within
the PBL (e.g., GEOS, GISS-MATRIX). For validation, we further compared the aerosol vertical profiles between models
and CALIOP observations (see **Figure 11b1-3**). To highlight the aerosol layer, we normalized each vertical profile based on
the maximum (EC_max) and minimum extinction coefficients (EC_min) to remove the magnitude difference. The
normalized EC was calculated as (EC_model – EC_min)/(EC_max-EC_min). Over AMAZ and SHAF, only a few models
(ECHAM-HAM, ECHAM-SALSA, CAM5-ATRAS, GISS-OMA, and GISS-MATRIX) could capture the peak aerosol
extinction at 2–4 km, whereas other models tended to show the strongest extinction at lower altitudes or the surface. Over
BONA, the observed extinction peaked at ~ 4 km, but no models were found with a similar profile. Compared with PEH
from CALIOP, the simulated BBA plume tended to be too low for all the models. A similar underestimation was also
reported elsewhere for AeroCom models with the bias being attributed to wet deposition being too strong in the models
(Koffi et al., 2016).



### 5.2 Diversity of MEC

Modeled MECs are affected by several factors (e.g., particle size, complex refractive index, and hygroscopicity). As BBA is dominated by OA and very similar refractive indices are used in models (see **Appendix**), the choice of refractive indices is not discussed. Here we mainly examined the impacts of particle size and hygroscopicity.

Because particle size information was missing for the AeroCom models, we used modeled AE as it is an indicator of particle size (Shuster et al., 2006). **Figure 12a** shows the dependence of modeled MECs on AEs. The modeled AE varied

from 0.21 to 2.2. Ambient particle size is the result of emitted particle sizes and particle processing after emission (see **Appendix**). Among all models, the lowest AE was found in INCA from CTRL16 due to the large contribution from coarse-mode SS, which also led to lower extinctions because of the lower MECs for SS than OA. When omitting INCA, a significant negative correlation was found between MECs and AEs ($r$ = -0.58), although there were large variations between models. The correlation for CTRL19 models that were driven by the same emission inventory was even stronger ($r$ = -0.73).

The negative correlation suggested that a larger size (smaller AE) resulted in a stronger extinction per mass unit for typical BB aerosols, which agreed well with the observations (Laing et al., 2016; Kleinman et al., 2020). This can also be explained by the Mie-scattering theory. In **Figure 12a**, we show the relation between MECs and AEs for pure OA aerosol based on the Mie-scattering theory. We assumed that the radius of dry OA particle ranged from 0.02 to 0.5 μm. The lower edge of the radius corresponded to the smallest emitted particle assumed in all the models examined (see **Appendix**) and the latter

indicated the upper edge of accumulation mode (Tegen et al., 2019). Hygroscopic growth was considered to occur based on the Kappa-Köhler theory under an RH of 50% and the kappa value for OA was set as 0.06 referring to Zhang et al (2012). A series of sensitive tests suggested that hygroscopic growth did not affect the calculation much. The refractive index was set to 1.53-0.0055$i$ as assumed in most models. The extinction cross-section was retrieved from the look-up table from ECHAM-HAM, based on which MECs and AEs were calculated. The calculated MEC increased with increasing particle

size (decreasing AE), which agreed with the modeled relations.

The negative correlation between AEs and MECs suggested the possibility of evaluating and subsequently constraining MEC by AE. In **Figure 12b** we validated modeled AEs against observations from POLDER-GRASP for the fire season. Because most of the AE data for CTRL19 models had a monthly resolution, we collocated all the model data with observations on monthly basis. Compared with POLDER-GRASP observations, the majority of models tended to

overestimate AEs by up to 0.85. BONA had the highest overestimation on average (0.27), followed by AMAZ and SHAF. Given the previous analysis of the MEC dependence on AE, the underestimation of particle size may have led to a considerable underestimation of MECs and thereby AOD. Similar to the impacts of precipitation, no strong correlation was found for AOD biases with AE biases, which was largely due to the interaction of multiple factors and non-linear model response.



The hygroscopicity was quantified as the extinction enhancement factor (EEF) which was defined as the ratio of AOD at the ambient relative humidity (RH) to AOD at zero RH (dry AOD). **Figure 13** illustrates the relation between MECs and EEFs in models. For most models, a small EEF ($< 2$) was observed, and we did not observe clear patterns between EEFs and MECs, probably because the hygroscopic growth was not significant given the low hydrophilicity of OA and the dry air condition during fire seasons. For comparison, we also collected the EEF for BBA from in-situ measurements (see **Table 3**).

The observed EEF ranges from 1 to 2.1, which was consistent with the values calculated by most models. Note that the 'dry' condition (RH = 20%−30%) in field measurements was different from that in models which assumed an RH of 0. The ambient RH in models over the three regions (47%−75%) also differed a lot from the RH in observations. However, given that BBA was dominated by OA and BC which were moderately hygroscopic or hydrophobic, the difference in the reference RH to calculate EEF might have small impacts (Tito et al., 2016). Apart from these models, there were a few models that

showed pretty strong hygroscopic growth, accompanied by a positive correlation with MECs for each model. Since EEF was related to the ambient RH and composition of hydrophilic aerosols (i.e., SU and SS), we further compared these values between the two groups of models with either low or high hygroscopicity. Both the modeled RH (averaged for layers from the surface to 650 hPa) and the percentage of hydrophilic aerosols show much smaller differences in the same fire regions between the two model groups compared with the large disparities in EEF. This suggested that the difference of EEF in the

two model groups was linked to the BBA properties, which might result from the modeled particle size, mixing state, and hygroscopicity parameterizations for OA (Burgos et al., 2020). For example, SPRINTARS assumed a similarly strong hygroscopicity for BBA compared with SU (Takemura, 2005). In addition, we also found the modeled relations between MEC and EEF were closely related to the treatment of 'clear-sky/all-sky' assumptions. For example, the clear-sky data from GISS-OMA model showed similar EEFs to other models, whereas the all-sky data exhibited much higher EEFs. For those

models with higher hygroscopicity (i.e., GEOS-Chem, GISS-OMA, IMPACT, and SPRINTARS), the predicted MECs under the same EEF varied substantially by a factor of 5 per model, suggesting that the modeled MEC diversity was controlled by other factors. When all models were considered together, there was no clear pattern between EEFs and MECs found.

**6. Conclusions**

In this paper, we conducted a comprehensive evaluation and interpretation of AOD errors in AeroCom models over three

key BB regions. We first evaluated 14 satellite AOD datasets against AERONET and identified their errors. These errors in satellite observations were then compared with model errors, with a much larger magnitude for the latter found in most models. We noticed that such a small impact from different satellite products only applied for our validations over BB





regions during fire seasons. Specially, we found that the errors due to different satellite observations were comparable to the model errors for the non-fire seasons over the three BB regions.

Detailed model validations against POLDER-GRASP observations suggested that most of the models still largely underestimated AOD, especially when using the standard emission inventories (e.g., GFED3, CMIP6). We did not observe significant improvements of modeled AOD in the latest experiment (CTRL19) compared with previous ones. The model ensembles from the three AeroCom experiments exhibited a smaller inter-model spread of AOD correlation with observations than AOD errors (e.g., CRMSE, NMB). Models seem to have a similar capability to model the spatiotemporal

variation of BBA, probably due to the similarity of input emissions as we found pretty strong correlations (~ 0.7) among the emission inventories used by these models (see **Appendix**). Most of the diversity in model errors is due to a season-wide bias. That said, temporal biases seem larger than spatial biases. We also provided evidence that AOD errors during the fire season were dominated by BBA errors, with only a small contribution from the background. Based on BBE simulations, we found negative biases could be reduced by scaling up BBA emissions. However, we showed that simulations with scaled

emissions did not thoroughly increase model performance.

We further analyzed the large diversity in fire AOD as resulting from emissions, lifetimes, and MECs which all exhibited large diversities too. When all models were considered, we showed that the contributions of these three factors to the overall AOD diversities were similar, though emissions exhibited slightly higher importance. In spite of the large inter-model diversities, individual models show very similar lifetime and MEC over different BB regions, suggesting that basic model

assumptions underlie lifetime and MEC for each model. We suspect that relatively simple changes in these assumptions may produce significant improvement in BBA simulations.

Modeled lifetime was correlated with modeled precipitation strength. Comparisons with observations suggested diverse and region-specific precipitation errors. Modeled lifetime was also related to plume height which was found to be strongly underestimated by models. We found MECs depended on how models simulate AE (or particle size). We further compared

modeled AE with POLDER-GRASP observations where general AE overestimations were found in most models. Most models produced acceptable hygroscopicity compared with observations. These findings can provide useful information for future model improvement and development.

There are several uncertainties in our evaluation and analysis. One is the uncertainties in POLDER-GRASP satellite observation. Although we showed that satellite errors did not affect our evaluations very much, we still found that POLDER-

GRASP had un-ignorable retrieval errors over the focused regions (13%). However, the retrieval error was difficult to be precisely defined due to the lack of sufficient samplings in SHAF and BONA by AERONET. On a global scale, POLDER-GRASP was found to be superior to other satellite products used in this study. The other uncertainty stems from the assumption of clear-sky conditions. As we evaluate model AOD against satellite data which are always clear-sky



observations, clear-sky model AOD should be used for comparison. However, models have very different treatments of the
'clear-sky' assumption (see **Appendix**). Although strict collocation can partly address this issue, uncertainties may still exist. Such an issue should be investigated more in further model validations.

**Data availability**

All the model data can be accessed at AeroCom Wiki (https://aerocom.met.no). POLDER-GRASP dataset can be found at https://download.grasp-cloud.com/download/polder/. All the other observations can be found in their references as listed.
The data processing in this work was done via CIS (http://www.cistools.net/). Codes to create individual figures can be obtained from the corresponding author upon request (q.zhong@vu.nl).

**Competing interests**

The authors declare that they have no conflict of interest.

**Acknowledgements**

This work was financially supported by Netherlands Organization for Scientific Research (NWO; ALWGO.2018.052). K.T. and S.E.B. acknowledge NASA MAP for support. Resources supporting this work were provided by the NASA High-End Computing (HEC) Program through the NASA Center for Climate Simulation (NCCS) at Goddard Space Flight Center. H.M. was supported by the Ministry of Education, Culture, Sports, Science and Technology of Japan and the Japan Society
for the Promotion of Science (MEXT/JSPS) KAKENHI Grant Numbers JP19H04253, JP19H05699, JP19KK0265, JP20H00196, and JP20H00638, MEXT Arctic Challenge for Sustainability phase II (ArCS-II; JPMXD1420318865) project, and the Environment Research and Technology Development Fund 2–2003 (JPMEERF20202003) of the Environmental Restoration and Conservation Agency. We thank all the modelers that have submitted AeroCom model data used in this work.




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





**Table 1. The details of the AeroCom Phase-III models evaluated in this study.**

| Model | AeroCom experiment[a] | | | Lat./Lon./Lev. | Meteorology | Reference |
|---|---|---|---|---|---|---|
| | BBE | CTRL16 | CTRL19 | | | |
| CAM-Oslo (CAM-Nor) | | ✓ | ✓ | 192×288×30 (CTRL16) 192×288×32 (CTRL19) | ERA-Interim | Kirkevåg et al, 2018; Seland et al., 2020 |
| CAM5 | ✓ | ✓ | | 96×144×30 | ERA-Interim | Liu et al., 2012 |
| CAM5-ATRAS | | | ✓ | 96×144×30 | MERRA2 reanalysis | Matsui, 2017; Matsui and Mahowald, 2017 |
| ECHAM-HAM | ✓ | ✓ | ✓ | 96×192×47 | ERA-Interim | Tegen et al., 2019 |
| ECHAM-SALSA | ✓ | ✓ | ✓ | 96×192×47 | ERA-Interim | Kokkola et al., 2018 |
| ECMWF-IFS | ✓ | ✓ | ✓ | 256×512×60 | ECMWF IFS forecasts | Rémy et al., 2019 |
| EMEP | | ✓ | ✓ | 360×720×20 | ECMWF IFS forecasts | EMEP, 2012 |
| GEOS | | | ✓ | 181×360×72 | MERRA-2 reanalysis | Colarco et al., 2010 |
| GEOS-Chem | ✓ | ✓ | ✓ | 46×72×47 (BBE) 91×144×47 (CTRL16) | MERRA-2 reanalysis | Bey et al., 2001 |
| GFDL | ✓ | | | 180×360×33 | NCEP/NCAR reanalysis | Donner et al., 2011 |
| GISS-MATRIX | ✓ | | ✓ | 90×144×40 | NCEP/NCAR reanalysis | Bauer et al., 2008 |
| GISS-OMA | ✓ | | ✓ | 90×144×40 | NCEP/NCAR reanalysis | Bauer et al., 2020 |
| HadGEM3 | | ✓ | | 144×192×38 | ERA-Interim | Bellouin et al., 2013; Mulcahy et al., 2020 |
| IMPACT | | ✓ | | 96×144×30 | | Liu et al., 2005 |
| INCA | ✓ | ✓ | | 143×144×79 | ECMWF | Balkanski et al., 2004; Schulz et al., 2009 |
| OsloCTM | ✓ | ✓ | ✓ | 64×128×60 (BBE) 80×160×60 (CTRL16/19) | ECMWF | Myhre et al., 2007; 2009 |
| SPRINTARS | ✓ | ✓ | ✓ | 320×640×40 | ERA-Interim (BBE, CTRL16) ERA5 (CTRL19) | Takemura et al., 2005 |
| TM5 | | ✓ | ✓ | 90×120×34 | ERA-Interim | van Noije et al., 2014; 2021 |

a. Note that different versions of models have participated in the 3 experiments.





**Table 2. Details of the satellite datasets used in this study.**

| Platform | Instrument | Algorithms/products | Dataset name | Reference |
|---|---|---|---|---|
| **Aqua** | MODIS | BAR v1.0 | Aqua-MODIS-BAR | Lipponen et al., 2018 |
| | | Deep Blue C6.1 | Aqua-MODIS-DB | Hsu et al., 2013; 2019; Sayer et al., 2019 |
| | | Dark Target C6.1 | Aqua-MODIS-DT | Remer et al., 2005 |
| | | MAIAC v2.0 | Aqua-MODIS-MAIAC | Lyapustin et al., 2018 |
| **Terra** | MODIS | BAR v1.0 | Terra-MODIS-BAR | Lipponen et al., 2018 |
| | | Deep Blue C6.1 | Terra-MODIS-DB | Hsu et al., 2013; 2019; Sayer et al., 2019 |
| | | Dark Target C6.1 | Terra-MODIS-DT | Remer et al., 2005 |
| | | MAIAC v2.0 | Terra-MODIS-MAIAC | Lyapustin et al., 2018 |
| **ENVISAT** | AATSR | ADV/ASV v2.30 | AATSR-ADV | Sogacheva et al., 2017 |
| | | ORAC v3.20 | AATSR-ORAC | Thomas et al., 2009 |
| | | SU v4.21 | AATSR-SU | North et al., 1999; North, 2002; Bevan et al., 2012 |
| **noaa18** | AVHRR | Deep Blue | AVHRR-DB | Sayer et al., 2017 |
| **SeaSTAR** | SeaWiFS | Deep Blue | SeaWiFS-DB | Hus et al., 2013 |
| **PARASOL** | POLDER | GRASP v2.1 | POLDER-GRASP | Dubovik et al., 2011 |



**Table 3. The extinction enhancement factor (EEF) for BBA at 550 nm wavelength from in-situ measurements.**

| Region | Dry RH, % | Reference RH, % | EEF | Reference |
|---|---|---|---|---|
| Brazil | 30 | 80 | 1.01−1.51 | Kotchenruther and Hobbs, 1998 |
| Australia | 20 | 80 | 1.1−1.7 | Gras et al., 1999 |
| Indonesia | 20 | 80 | 1.2−2.1 | Gras et al., 1999 |
| Southern Africa | 30 | 80 | 1.66 ± 0.08 (fresh) 1.42 ± 0.05 (aged) | Magi and Hobbs, 2003 |
| India | 40 | 85 | 1.58 ± 0.21 | Sheridan et al., 2002 |
| China | 30 | 80 | 1.64 | Jung and Kim, 2011 |
| India | ≤40 | 85 | 1.32 ± 0.14 | Dumka et al., 2017 |


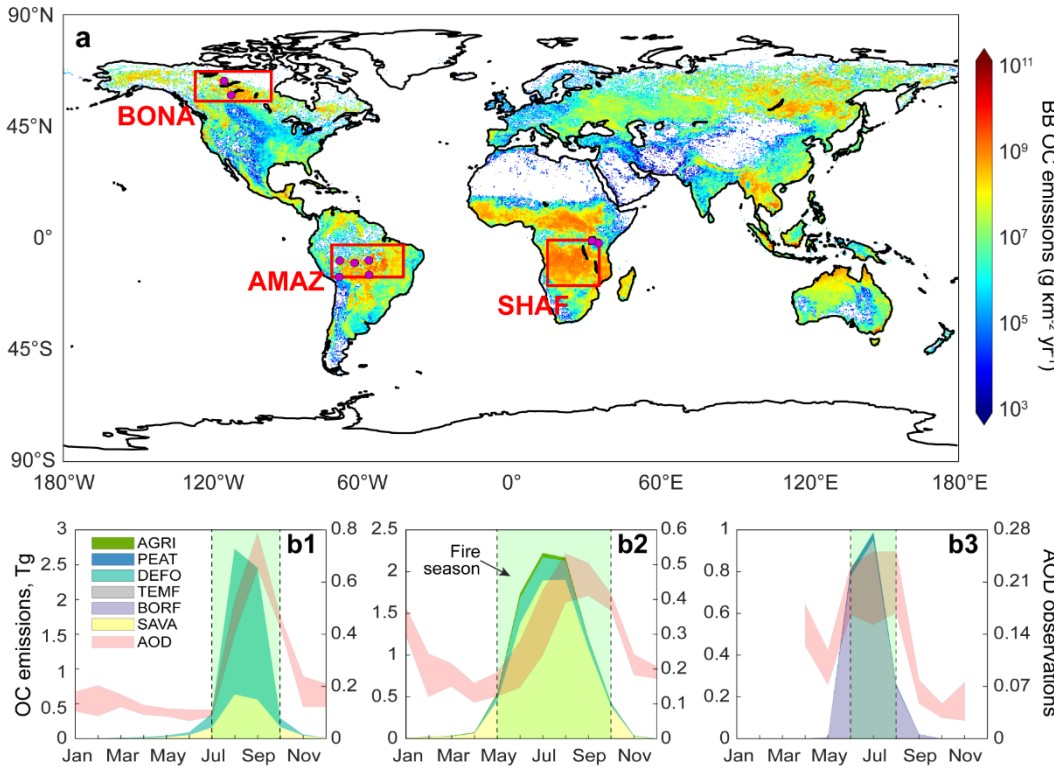

**Figure 1. Three focused fire regions in this study.** a) Global maps of BB OC emissions averaged for 2006, 2008, and 2010 based on GFED4.1s (https://www.globalfiredata.org/). The domains of the three fire regions are shown by the red boxes together with the AERONET sites (purple dots). b) The monthly evolutions of BB OC emissions from six fire types in AMAZ (b1), SHAF (b2), and BONA (b3), respectively. The un-collocated regional mean AOD observations from 14 satellite datasets are shown by the light-red shaded areas as inter-quartile ranges (only the grid-boxes with more than 20 data available in a month are included). Emissions for BB were considered in terms of the biome/fire type: tropical forest and deforestation (DEFO), savanna (SAVA), temperate forest (TEMF), boreal forest (BORF), peat (PEAT), and agricultural waste (AGRI).





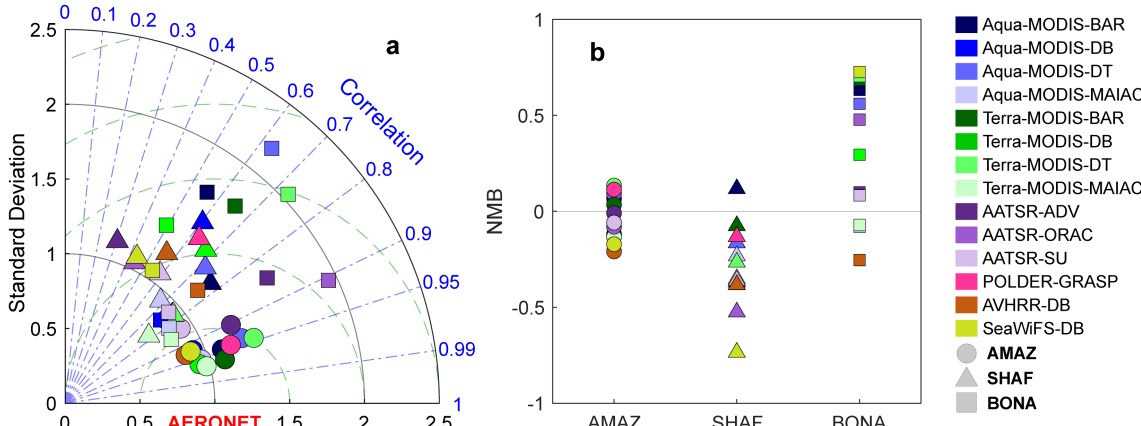

**Figure 2. Comparison of the 14 satellite AOD products against AERONET observations as shown by a Taylor diagram (a) and scattering diagram of NMB (b).** The colors and shapes of dots indicate different satellite datasets and fire regions. All the satellite data were individually collocated with AERONET data during the fire seasons. POLDER-GRASP and SeaWiFS products over BONA are not shown because the available sample size was too small (< 5) after collocation.





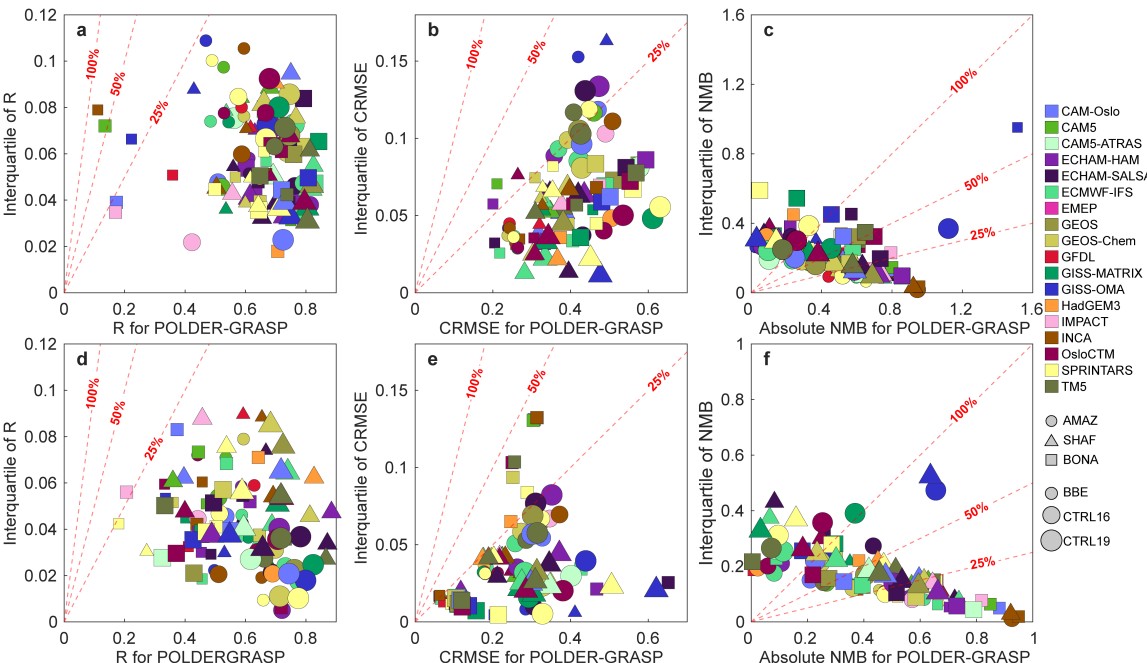

**Figure 3. Variation of model AOD evaluation due to different choices of satellite products in terms of the correlation coefficient (a, d), centered root mean square error (b, e), and normalized mean bias (c, f).** The results are shown as comparisons between the values using POLDER-GRASP (horizontal axis) and interquartile ranges (vertical axis) when validating each model with different satellite products. The top (a-c) and bottom panels (d-f) show the results for individual and synchronous collocation, respectively. The color, shape, and size of dots indicate different models, three fire regions, and three AeroCom experiments respectively. The dashed lines show the 25%, 50%, and 100% slopes (interquartile/median). The GISS-OMA data for BBE over BONA is not shown in (b) due to the very high CRMSE.



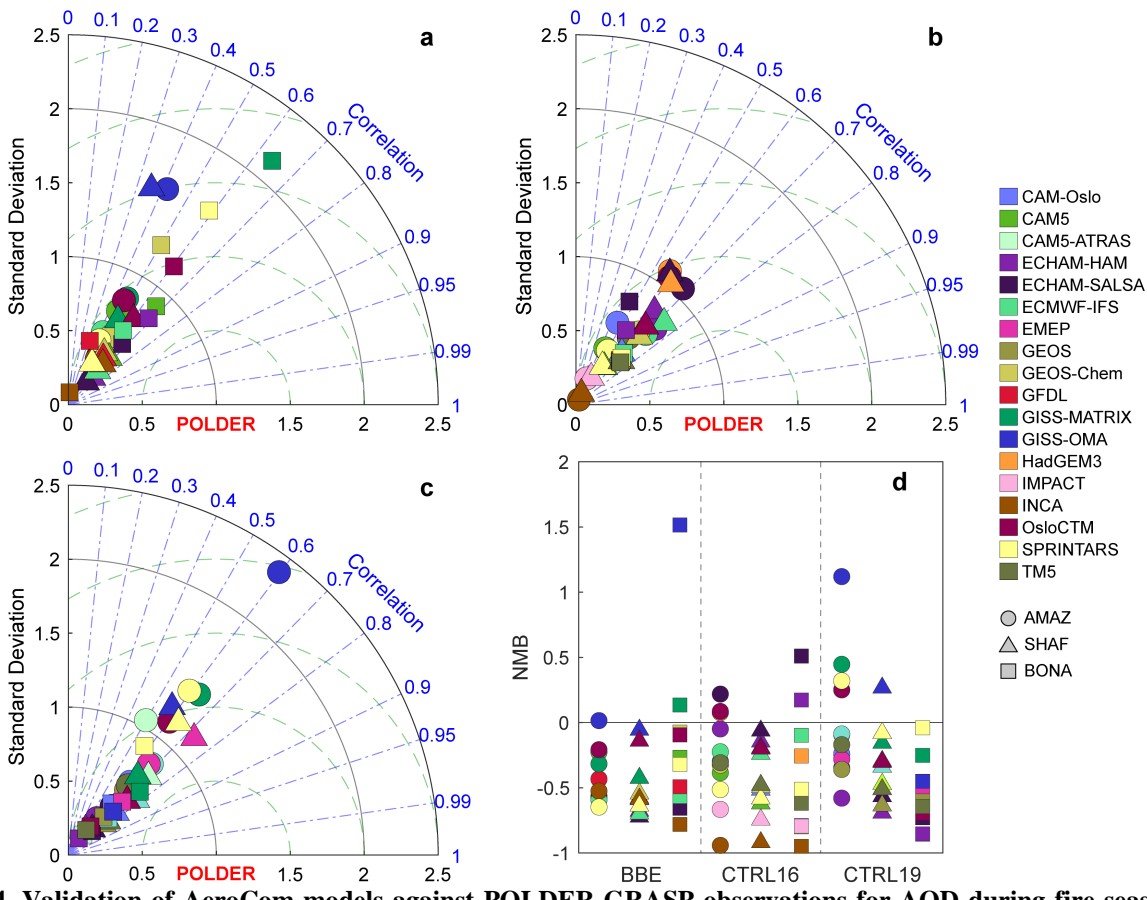

**Figure 4. Validation of AeroCom models against POLDER-GRASP observations for AOD during fire seasons.** The validations of models from the three AeroCom experiments are shown as Taylor diagrams for BBE (a), CTRL16 (b), and CTRL19 (c). The NMB for all the models is shown in panel d. The colors and shapes of dots indicate different models and fire regions. All the model data are collocated with POLDER-GRASP data. The evaluation is for 2008 in BBE, for 2006, 2008, and 2010 in CTRL16, and for 2010 in CTRL19. The GISS-OMA data for BBE over BONA is not shown in (a) due to the very large normalized CRMSE.





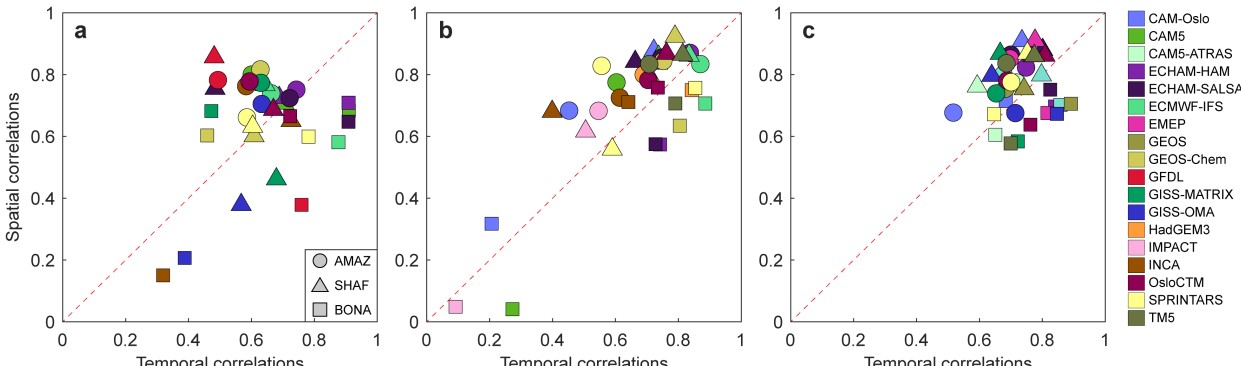

**Figure 5. Comparison of the temporal and spatial correlations between modeled AOD and POLDER-GRASP observations.** Results were shown for the three experiments individually (a-BBE, b-CTRL16, c-CTRL19). All the model data are collocated with POLDER-GRASP during fire seasons. The correlations are then calculated using either the time-series of the regional averages (horizontal axis) or the spatial averages for the whole fire seasons (vertical axis). The red dashed lines show the 1:1 range.





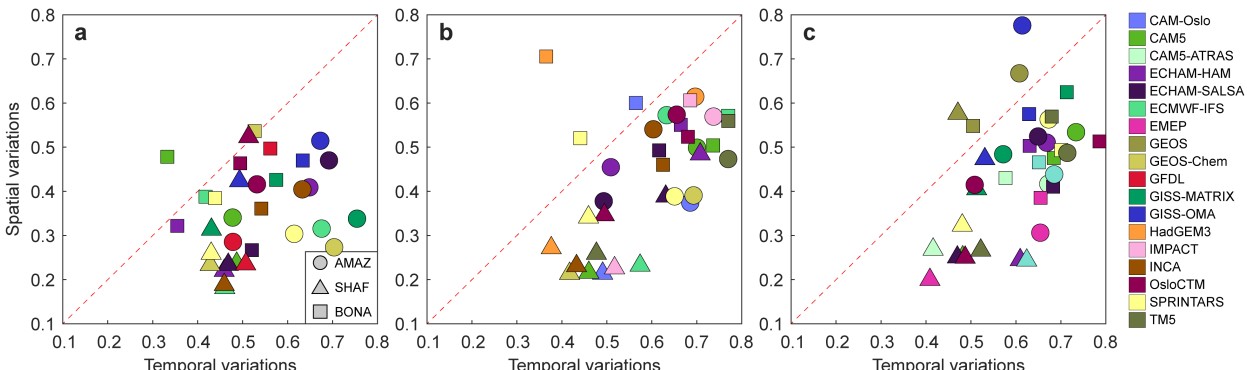

**Figure 6. Comparison of the temporal and spatial variations of modeled AOD errors.** Results are shown for the three experiments individually (a-BBE, b-CTRL16, c-CTRL19). All the model data were collocated with POLDER-GRASP. The variation is calculated as the ratio of interquartile to median values of the absolute bias for time series (temporal variations) and spatial averages (spatial variations). The red dashed lines show the 1:1 range.





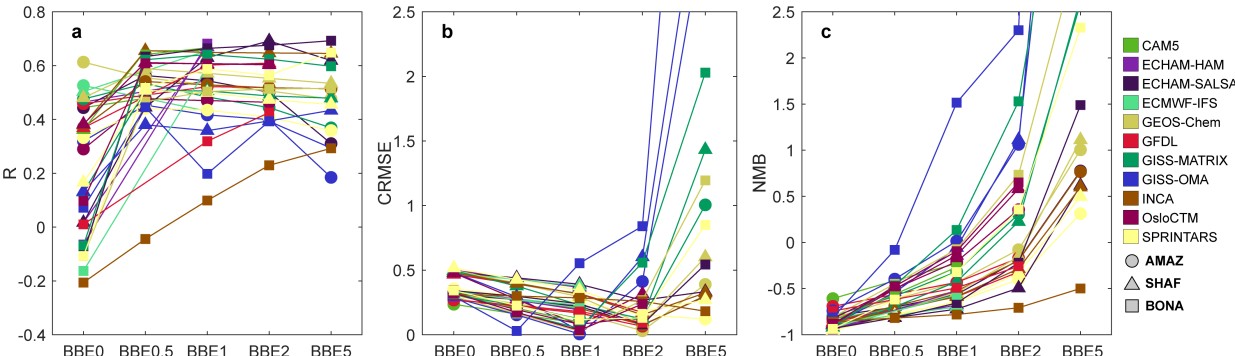

**Figure 7. Changes of correlation (a), centered root mean square error (b), and normalized mean bias (c) in the BBE experiment in responding to different scaling factors adopted to BBA emissions (0, 0.5, 2, 5).** The colors and shapes of dots indicate different models and fire regions. All data are collocated with POLDER-GRASP for 2008 fire seasons. The BBE5 CRMSE and NMB for several models are not shown given the extremely large values (up to 19).





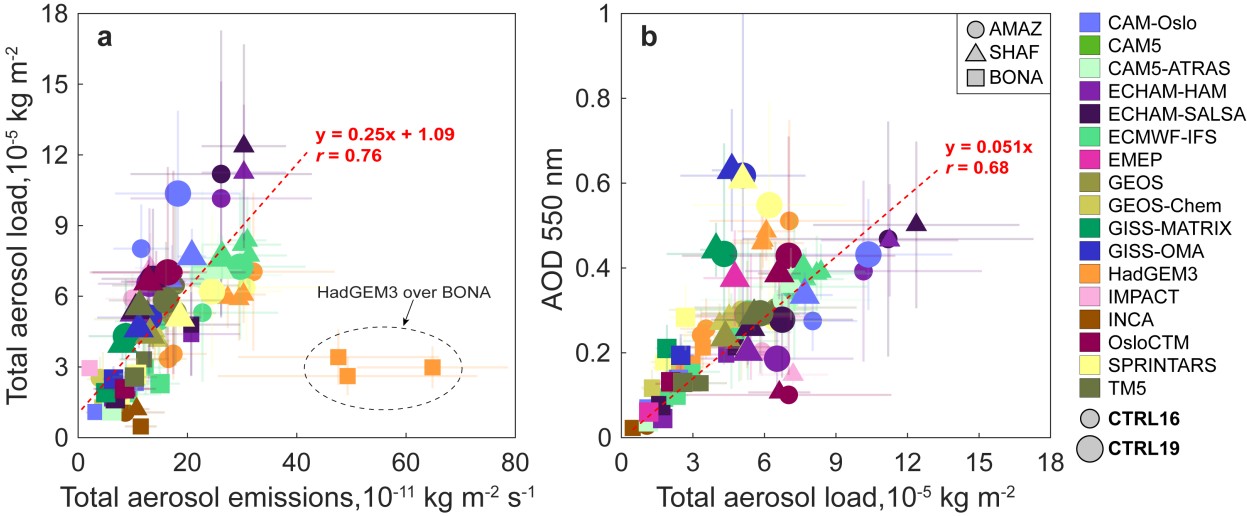

**Figure 8. The dependence of total aerosol load on total aerosol emissions (a) and dependence of AOD on aerosol load (b), indicating the aerosol lifetime and MECs, respectively.** The data are average values for the whole fire seasons based on the raw model output without collocation, and the light-colored error bars indicate the corresponding temporal variations (as standard deviation). The color, shape, and size of dots indicate different models, three fire regions, and two AeroCom control experiments, respectively. The red dashed lines show the linear trends, with a regression function and correlation coefficients (*r*) also shown. Note that some CTRL16 models provide data for different years (2006, 2008, and 2010) which are illustrated separately.





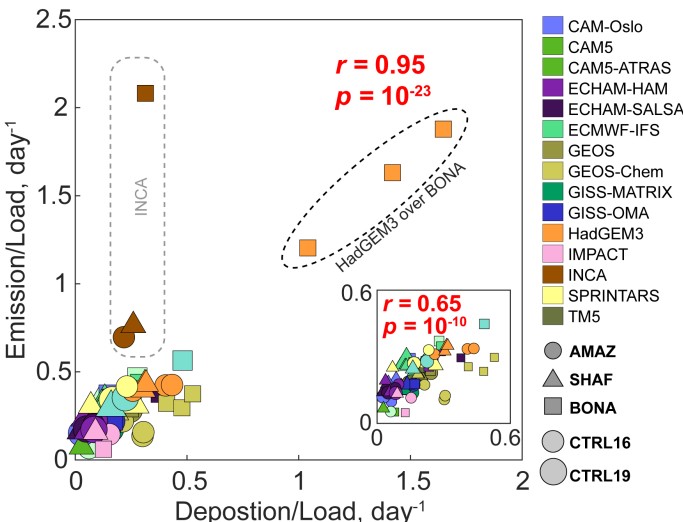

**Figure 9. Dependence of the modeled aerosol lifetime on the time scale of total deposition.** The color, shape, and size of dots indicate different models, three fire regions, and two AeroCom control experiments, respectively. The embedded diagram shows the same results zooming to a smaller scale (excluding INCA and HadGEM3 in BONA). The Person correlation (*r*) and *p*-value (*p*) are shown.



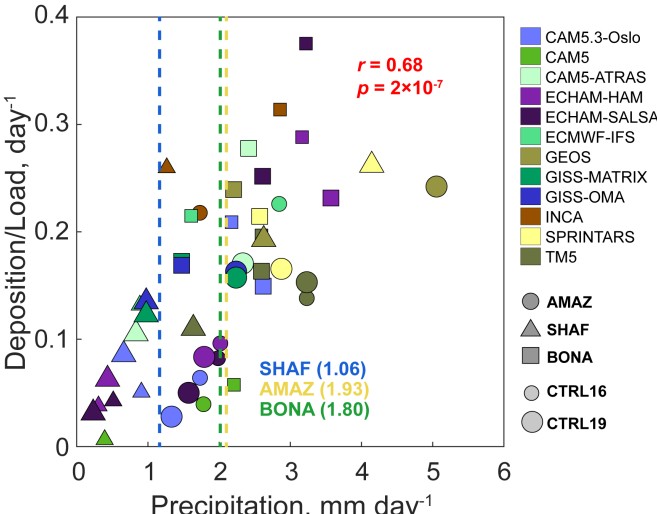

**Figure 10. Dependence of the modeled timescale of total deposition on precipitation strength during fire season in 2010.** The color, shape, and size of dots indicate different models, three fire regions, and two AeroCom control experiments, respectively. The three dashed lines indicate the GPCP data averaged for each region.


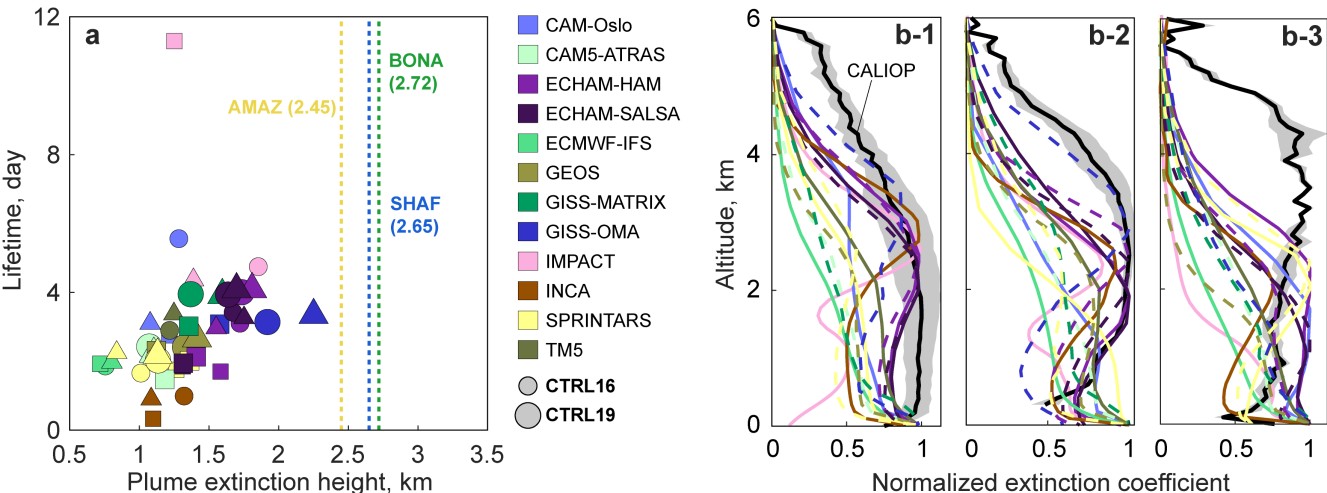

**Figure 11. Variation in modeled plume height (a) and validation of modeled aerosol vertical profile against CALIOP for AMAZ (b-1), SHAF (b-2), BONA (b-3).** The color scheme in **Fig. 11b1-3** is the same as in **Fig. 11a**, with the solid and dashed lines showing the model data from CTRL16 and CTRL19 experiments (if both are available for the same model), respectively. The gray shaded areas in Fig. 11b show the ±σ ranges for the CALIOP observation.





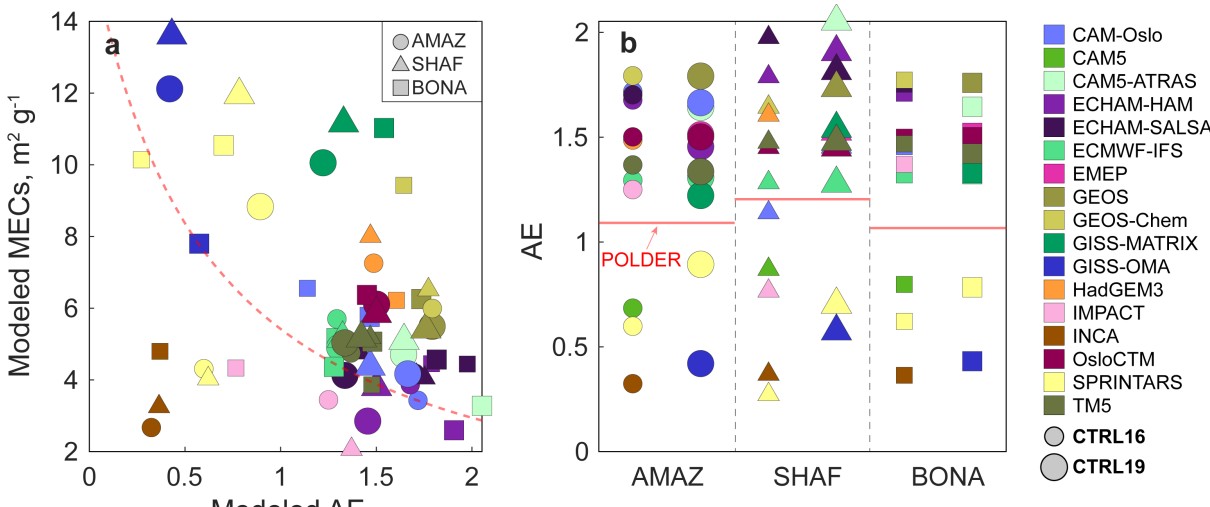

**Figure 12. Dependence of modeled MECs on AE (a) and the validation of modeled AE against POLDER-GRASP data (b).** Data in **Fig. 12a** are model original output without collocation. The dashed line in **Fig. 12a** shows the relation calculated based on Mie-scattering theory and ECHAM-HAM lookup table. Modeled AE in **Fig. 12b** are collocated with POLDER-GRASP (shown as red lines) on monthly basis during fire seasons.



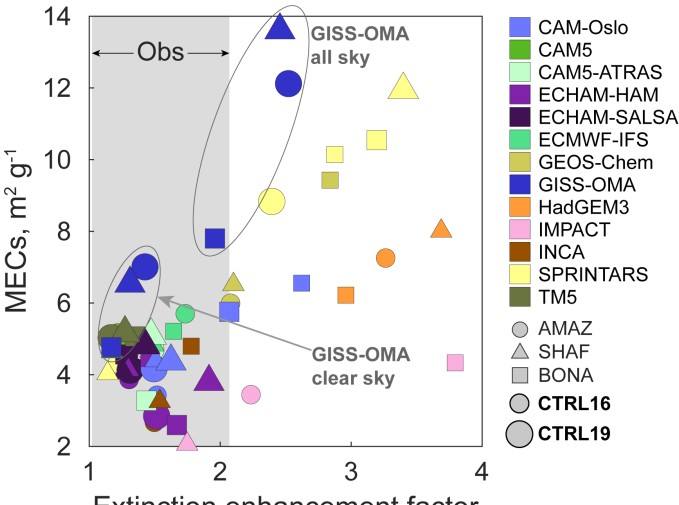

**Figure 13. Dependence of modeled MECs on extinction enhancement factor (EEF) in models for 2010.** The grey shaded area shows the EEF range from in-situ observations according to previous studies (see **Table 3**). Both clear-sky and all-sky results are shown for GISS-OMA data.