# Peer review of "Satellite-based evaluation of AeroCom model bias in biomass burning regions"

_Atmospheric Chemistry and Physics, 2022_

## Author Comment (AC1)

**Response to Referee #1 on acp-2022-96**

**Review of Zhong et al, Satellite-based evaluation of AeroCom model bias in biomass burning regions.**

This paper present an evaluation of AeroCom model aerosol optical properties in regions strongly influenced by biomass burning. In line with previous research, large biases are found. Diverse satellite products are used and a valuable comparison of satellite products is included. Furthermore, a useful disentangling of the biases associated with emissions and with lifetime is presented. The paper is well written and has potential to be an important contribution to ACP. I have a number of minor comments which should be addressed before the paper is published.

**Response**

We sincerely thank the reviewer for the valuable comments. These comments have been carefully addressed during revision. Please find our point-to-point response below and highlighted changes in the revised manuscript.

**Minor comments**

Models and variables section: More historical context on when the simulations were run and what the differences in model versions are between the experiments would be useful here. Were the model versions the same, or did the models change between the BBE, 2016 and 2019 experiments? I don't think you can expect your readers to be familiar with AeroCom protocols or to go through other AeroCom papers or the Excel sheet supplement, though of course all the details of specific changes from one experiment to the next do not need to be repeated here.

**Response**

Thank you for the suggestion. The simulation period was stated for each experiment (please see lines 137, 147, 152). Regarding the details of models, we have added the model version information to the updated Table 1. The following sentence were added to provide a better reference.

**Lines 132-134**: "A total of 18 different models were investigated in our study, with parts of the models participating in multiple experiments with different versions. **Table 1** provides an overview of these models, more details are provided in the **Appendix** and listed references."

**Table 1. The details of the AeroCom Phase-III models evaluated in this study.**

| Model | Experiment[a] | Model version | Lat./Lon./Lev. | Meteo. | Reference |
|---|---|---|---|---|---|
| CAM-Oslo | CTRL16 | CAM5.3-Oslo | 192×288×30 | ERA-Interim | Kirkevåg et al, 2018; Seland et al., 2020 |
| | CTRL19 | NorESM2 (CAM6-Nor) | 192×288×32 | | |
| CAM5 | BBE | CAM5.3_f19 | 96×144×30 | ERA-Interim | Liu et al., 2012 |
| | CTRL16 | CAM5.3_f19 | | | |
| CAM5-ATRAS | CTRL19 | CAM5-ATRAS-v2.0 | 96×144×30 | MERRA2 | Matsui, 2017; Matsui and Mahowald, 2017 |
| ECHAM-HAM | BBE | ECHAM6.1-HAM2.2 | 96×192×47 | ERA-Interim | Tegen et al., 2019 |
| | CTRL16 | ECHAM6.3-HAM2.3 | | | |
| | CTRL19 | ECHAM6.3-HAM2.3 | | | |
| ECHAM-SALSA | BBE | ECHAM6-SALSA | 96×192×47 | ERA-Interim | Kokkola et al., 2018 |
| | CTRL16 | ECHAM6-SALSA | | | |
| | CTRL19 | ECHAM6.3-SALSA2.0 | | | |
| ECMWF-IFS | BBE | ECMWF-IFS-CY45R1-CAMS | 256×512×60 | ECMWF-IFS | Rémy et al., 2019 |
| | CTRL16 | ECMWF-IFS-CY42R1-CAMS | | | |
| | CTRL19 | ECMWF-IFS-CY46R1-CAMS | | | |
| EMEP | CTRL19 | EMEP_rv4_33 | 360×720×20 | ECMWF-IFS | EMEP, 2012 |
| GEOS | CTRL19 | GEOS-i33p2 | 181×360×72 | MERRA-2 | Colarco et al., 2010 |
| GEOS-Chem | BBE | GEOS-Chem-v9-02 | 46×72×47 | MERRA-2 | Bey et al., 2001 |
| | CTRL16 | GEOS-Chem-v11-01 | 91×144×47 | | |
| GFDL | BBE | GFDL-AM4p0 | 180×360×33 | NCEP/NCAR | Donner et al., 2011 |
| GISS-MATRIX | BBE | GISS-ModelE2-MATRIX | 90×144×40 | NCEP/NCAR | Bauer et al., 2008 |
| | CTRL19 | GISS-ModelE2p1p1-MATRIX | | | |
| GISS-OMA | BBE | GISS-ModelE2-OMA | 90×144×40 | NCEP/NCAR | Bauer et al., 2020 |
| | CTRL19 | GISS-ModelE2p1p1-OMA | | | |
| HadGEM3 | CTRL16 | HadGEM3-GA7.1 | 144×192×38 | ERA-Interim | Bellouin et al., 2013; Mulcahy et al., 2020 |
| IMPACT | CTRL16 | IMPACT | 96×144×30 | | Liu et al., 2005 |
| INCA | BBE | INCA | 143×144×79 | ECMWF | Balkanski et al., 2004; Schulz et al., 2009 |
| | CTRL16 | INCA-BCext | | | |
| OsloCTM | BBE | OsloCTM2 | 64×128×60 | ECMWF | Myhre et al., 2007; 2009 |
| | CTRL16 | OsloCTM3 | 80×160×60 | | |
| | CTRL19 | OsloCTM3v1.02 | 80×160×60 | | |
| SPRINTARS | BBE | MIROC5.2-SPRINTARS | 320×640×40 | ERA-Interim | Takemura et al., 2005 |
| | CTRL16 | MIROC5.9.0-SPRINTARS | | | |
| | CTRL19 | MIROC6-SPRINTARS | | ERA5 | |
| TM5 | CTRL16 | TM5-mp | 90×120×34 | ERA-Interim | van Noije et al., 2014; 2021 |
| | CTRL19 | TM5-mp-r1058 | | | |

**a.** Models participated in one or multiple experiments with either the same or different model versions. The experiments include biomass burning emission (BBE) for 2008, CTRL 2016 (CTRL16) for 2006/2008/2010, and CTRL 2019 (CTRL2019) experiments for 2010.

Even though the size distribution of the model output is not available, the size distributions of the simulated BB emissions inputs are mentioned in the Appendix Table, so it should be possible to infer the impact of these size distributions on lifetime and AOD to some extent. It would be useful to try to do this, and it seems odd to have such a long discussion on hygroscopicity when size is probably more important.

**Response**

Thank you for the suggestion. It would be indeed nice to have discussions on the impacts of emitted particle size distribution. However, we find the emitted particle size is not the only reason driving changes in ambient particle size (and subsequently in lifetime and MEC), since microphysics in different models (e.g., nucleation, coagulation, etc.) also plays a role. For example, GISS-MATRIX assumes a small size for BB emissions (diameter = 50 nm) but produces very low AE (and high MEC).

Practically, involving discussions regarding emitted size and model processes will significantly increase the size of the paper which is already large. We are currently working on another paper which focuses on the impacts of particle size. The current paper still concentrates on evaluating the bias and diversity of modeled AOD over fire regions.

We also re-wrote the paragraph regarding hygroscopic growth, please see lines 482-496.

Why does the NMB for BBE5 reach up to 19? Isn't it a bit surprising that it ever exceeds 7.5, given BBE1 has a maximum NMB of 1.5? Is this a linear increase (line 372)?

**Response**

Thank you for the note. The model with the highest NMB was over BONA by GISS-OMA. The non-linearity at high emissions stemmed mainly from two grid boxes with extremely high AOD values (> 20). Given the model was collocated with POLDER-GRASP satellite observation which had a small sampling size over BONA, these extremes led to an abnormally high mean value in BBE5. If we use a median value for regional AOD, we get a rather linear response (please see the following Fig. R1). However, this metric would lead to inconsistency to the overall paper where the normalized mean bias was mostly used. In the revised manuscript, we have removed the value in the caption of Fig. 8 to avoid confusion.

[Figure]

**Fig. R1** Modeled median AOD in response to different scaling factors of BBE experiment by GISS-OMA model.

Please note that the NMB is calculated as *AOD_mo/AOD_obs - 1*, so it is possible to get a NMB higher than 7.5 (since the default *AOD_mo/AOD_obs* = 2.5). The linear relationship in Fig. 9 is based on the whole model ensemble. An individual model can possibly deviate from such a trend.

Figure 5: I find this figure hard to extract much meaning from – a great deal of the information is lost by just showing charts of the correlations. I did not understand the value of a correlation between spatial correlation and temporal correlation. I think it would be better to have AOD vs time line plots for POLDER and for all of the models, with one subfigure for each region (or similar). Then we could see which part of the season the biases are most apparent in, and where the biases are in the regions. It is surprising the spatial correlation can be so low for some models (GISS and INCA) – perhaps a scatter plot would be useful here of simulated AOD vs POLDER AOD for these models?

**Response**

Thank you for the suggestions. The temporal correlation was calculated as the correlation coefficients between the modeled and observed time series of AOD averaged for the whole region. The spatial correlation referred to the correlation between model and observations for the grid-boxes with seasonal averaged AOD (please see lines 292-293). Here we would like to use these spatial and temporal correlations (and variations, please see Fig. 7) to investigate the driver of the overall correlations (and bias). We did not show the plots for original model

individually since they might be too massy to present sufficient information. Instead, we added a new plot showing the time-series of AOD for both POLDER-GRASP observations and the model ensemble values (please see Fig. 5).

The low correlations for GISS and INCA in BBE (Fig. 6a) were caused by several grid-boxes with extremely high AOD in the models. If omitting these grid-boxes, the correlations would agree with other models (0.6~0.8). The three models with low correlations in CTRL16 (Fig. 6b) were those using very different emissions (e.g., CMIP5 emissions for 2000 used by CAM-Oslo and IMPACT).

The following sentences were added regarding the new Fig. 5.

**Lines 290-291**: "In **Figure 5**, we compared the daily AOD series for the model ensembles with POLDER-GRASP observations. For most models, the underestimations of AOD tended to be exacerbated during the peak of AOD observations."

[Figure]

**Figure 5. Daily time-series for the AOD for AMAZ (left panels), SHAF (middle panels), and BONA (right panels) in BBE (a), CTRL16 (b), and CTRL19 (c) experiments.** All the model data were collocated with POLDER-GRASP during fire seasons. The model results are shown as model ensemble medians (solid lines) together with the interquartile ranges of the model spread (dashed lines). Data are averaged for 2008 in BBE, 2010 for CTRL16, and 2010 for CTRL2019, respectively.

Also, why are the results in subfigures a, b and c so different? What differed between the three experiments to cause this? You comment in the text that the figures are pretty similar, but they look quite different to me.

**Response**

Thank you for the note. The differences for the same model in different experiments are mainly driven by the input emissions. The GFED3 dataset was used in BBE and the CMIP6 dataset was used in CTRL19. Models in CTRL16 used their preferred emission inventories. Please see lines 136-152 for the clarification. Since diverse emission datasets were used in CTRL16, we found the models in this experiment showed the highest spread. In particular, the models that significantly deviated from others used very different emissions (e.g., CMIP5 emissions for 2000 used by CAM-Oslo and IMPACT, as shown in Fig. 6b).

Here we would like to conclude that even with the possible improvement of emission inventories (e.g., from GFED3 to CMIP6) and models (e.g., updated model versions), the correlations did not show significant improvement for the whole model ensemble (not for individual models which might be improved to a certain extent). This is supported by the t-test on models that participated in multiple experiments ($p > 0.05$). We have revised the relevant sentences as follows:

> **Lines 299-301**: "For the model ensembles, we found there was no significant difference among the three experiments for both spatial and temporal correlations, even though improvements in emission inventories and/or models might occur following the time sequence from BBE to CTRL16 to CTRL19."

L381-390 this is a nice analysis, should be very useful.

**Response**

Thank you for the support.

What is the real distinction between section 4 and section 5, before section 5.1? The sections may need more thought.

**Response**

In section 4, we collocated model data with POLDER-GRASP (i.e., sparsely distributed observation) to evaluate the modeled AOD bias from multiple aspects. In section 5, we mainly focused on understanding the drivers of model diversity which was based on the original model output (without collocations). Investigations on the model diversity provides a way to better understand (and potentially fix) the bias we have observed in section 4.

In revised manuscript, we added a new subtitle to the section before 5.1: '**Decomposition of modeled AOD diversity**'.

The following sentences were re-written to clarify the relation with previous section.

> **Lines 331-334**: "As the above model evaluation could not provide sufficient information on the causes of the model biases, we explore the diversities of AOD in this section. Our strategy is to first evaluate the diversity in modeled AOD and the possible drivers that could lead to such variability, and then compare those drivers with available observations to understand the model variability and therefore bias. This practice will also contribute to future model development."

L450-465: The interesting part here is not so much the negative correlation, which is presumably coded into the models by their parameterizations of Mie theory, but why the models deviate from the Mie curve- presumably due to the mixing of several broad size distributions.

**Response**

Thank you so much for the suggestion. We have added the following discussion on the possible reasons for the difference between models and our Mie calculations.

> **Lines 470-473**: "Note that many models deviate from our Mie calculation though the Mie theory was applied in those models. Possible causes for such deviations might include, e.g., the aerosol composition (i.e., non-OA components), mixing state for multiple species (e.g., BC), assumptions on the size distribution (e.g., bins, distribution width), and treatment for the mixing of particles with different size distributions."

L541 I did not see a discussion of the clear-sky assumption in the appendix, and the references given there are mostly generic model description papers, so it would take the reader unfeasibly long to reconstruct what difference the authors are referring to, so please clarify.

**Response**

Thank you for the suggestion. We added the following sentence to give an example of the model difference in determining 'clear-sky' conditions.

**Lines 534-536**: "However, models have very different treatments of the 'clear-sky' assumption. For example, SPRINTARS considers 20% cloud fraction as 'clear sky', while GISS-OMA assumes cloud free only for 0% cloudiness."

**Technical corrections**

**Abstract: "comprise" at line 60 is the wrong word**

**Response**

Thank you. It was revised to 'contain'.

**L240 "proposed" is an odd word here.**

**Response**

Thank you. It was revised to 'we utilized POLDER-GRASP to …'.

**L240-270 the paragraph is much too long and should be split up, with clearly defined topics introduced in the first sentence of each paragraph. That said, the paragraph from 271 to 274 does not have its own topic and seems to belong with the previous text.**

**Response**

Thank you for your suggestion. We have split the paragraph into 3 smaller ones. Please see lines 240-275. We also rewrote the previous paragraph from 271 to 274 to highlight the relation to previous contents (please see lines 272-275). In addition, we added subtitles to section 3 to make it more readable.

**L256 improve sentence**

**Response**

Thank you. The sentence was revised as follows:

**Lines 255-256**: "which suggested that different satellite products tended to have higher consistency in capturing the spatiotemporal variations than the magnitude of AOD."

**L520 not clear what 'thoroughly' means**

**Response**

Thank you. We revised the sentence as follows:

**Lines 513-515**: "However, we showed that scaling up emissions was not a perfect solution to address model bias as the correlations did not improve significantly, suggesting that the spatial and/or temporal bias still existed."

---

## Author Comment (AC2)

**Response to Referee #2 on acp-2022-96**

Dear Authors,

Thank you for this exhaustive and well-described analysis of the factors governing uncertainty in simulation of atmospheric aerosols in regions affected by biomass burning. This is a problem of long-standing concern in the atmospheric composition community, and your study provides valuable information on the commonalities and differences of the atmospheric simulation models currently in use.

I have only minor recommendations for revisions. I encourage you to also attend closely to the revisions requested by the other reviewers.

**Response**

We sincerely thank the reviewer for the overall support on our work. All the recommendations have been carefully addressed during revision. Please find our point-to-point response below and highlighted changes in the revised manuscript.

Line 226 The Schutgens (2020) paper makes a number of interesting assertions about the potential effects of cloud contamination, but I do not see the suggestion there that southern hemisphere Africa during the burning season is subject to high cloud contamination. That is not consistent with other literature either. I would examine other explanations such as the extent of arid areas in southern Africa where satellite retrieval is more difficult.

**Response**

Thank you for the suggestion. The difference is now interpreted as the higher surface reflectance in these less forested regions.

**Lines 224-225**: "probably due to the higher surface reflectance (less forested) which made the retrievals more difficult and less accurate (Fraser and Kaufman, 1985)."

**Ref**

Fraser, R. S., and Kaufman, Y. J.: The relative importance of aerosol scattering and absorption in remote sensing. IEEE Trans. Geosci. Remote Sens., 5, 625-633, https://doi.org/10.1109/TGRS.1985.289380, 1985.

Line 377: "For the aerosol lifetime and MEC which were mainly affected by other model aspects than emissions, there was no significant difference found among the three fire regions for the same model." Are you saying that the models used each had uniform MEC among the three regions? Are you saying that the models did not have varying lifetimes for the three regions? Either of these findings is quite significant, as they represent model assumptions and outcomes that can be compared to observations.

**Response**

Thank you for the note. Here we would like to state that the ensemble median values for lifetime and MEC do not differ much among the three fire regions. In particular, the ensemble median MECs over the three regions were lower than observations, indicating that models need to be improved. The similarity seems to suggest that we could modify some basic assumptions to improve model performance, which is carried out in our following work (in preparation).

For individual models, we did see differences in MEC and lifetime per region which was small for most models. The following sentences were re-written to clarify our finding.

**Lines 357-359**: "For the aerosol lifetime and MEC which were mainly affected by other model aspects than emissions, we found the ensemble median values for these two factors were similar among the three fire regions."

**Lines 518-520**: "In spite of the large inter-model diversities, the model ensembles show very similar lifetime and MEC over different BB regions, suggesting that basic model assumptions underlie lifetime and MEC for the current model ensemble."

Line 118: "regarding to knowing issues for BBA models for more than ten years" I would update this sentence and expand to clarify that BBA has been acknowledged as a large source of uncertainty in atmospheric aerosol for a very long time (e.g. AeroCom phase II paper from 2013: https://acp.copernicus.org/articles/13/1853/2013/,

or before that this 2005 review by Kanakidou https://acp.copernicus.org/articles/5/1053/2005/, or before that this 1992 Science paper by Joyce Penner https://www.science.org/doi/abs/10.1126/science.256.5062.1432), and this study was undertaken to examine uncertainties and variation in current state-of-the-art modeling systems.

**Response**

Thank you for the suggestion. We have revised the sentence as follows.

**Lines 116-118**: "The aim of this work is to provide a satellite-based assessment of the state-of-the-art global models in representing BBA that has long been recognized as an important contributor to the overall aerosol uncertainties (Kanakidou et al., 2005; Myhre et al., 2013)."

**Refs**

Kanakidou, M., et al., Organic aerosol and global climate modelling: a review, Atmos. Chem. Phys., 5, 1053-1123, https://doi.org/10.5194/acp-5-1053-2005, 2005.

Myhre, G., et al., Radiative forcing of the direct aerosol effect from AeroCom Phase II simulations, Atmos. Chem. Phys., 13, 1853–1877, https://doi.org/10.5194/acp-13-1853-2013, 2013.

Line 135 "in multi" => "in multiple"

**Response**

Thank you. Revised accordingly. Please see line 133.

Line 191 "To avoid sampling issues" => "To mitigate sampling issues associated with varying coverage of the observational data sources"

**Response**

Thank you. Revised accordingly. Please see line 190.

Line 256: "impacts of different were" "impacts of verifying against different satellite data products were"

**Response**

Thank you. Revised accordingly. Please see lines 256-257.

Line 270: "for the whole research" => "for the whole analysis."

**Response**

Thank you. Revised accordingly. Please see line 271.

Line 421: "positive correlation" is this actually a positive correlation? Your figure shows a positive correlation between precip and deposition load.

**Response**

Thank you for the note. The correlation is between precipitation and the deposition timescale (i.e., deposition/load). We have added such an item to the text. Please see line 410.

Line 520: "thoroughly" choose a different word—perhaps you mean "uniformly?"

**Response**

Thank you. We revised the sentence as follows to make it clear:

**Lines 513-515**: "However, we showed that scaling up emissions was not a perfect solution to address model bias as the correlations did not improve significantly, suggesting that the spatial and/or temporal bias still existed."

---

## Author Response (AR2)

Dear Editor,

Thank you so much for accepting our paper. We appreciate your time and efforts very much. Here we submitted all the required files for production. Should you have any questions, please feel free to contact us.

Sincerely yours,

Qirui Zhong
On behalf of all authors